# Application of Optical Communication for an Enhanced Health and Safety System in Underground Mine

**DOI:** 10.3390/s23020692

**Published:** 2023-01-07

**Authors:** Olimpiu Stoicuta, Simona Riurean, Sorin Burian, Monica Leba, Andreea Ionica

**Affiliations:** 1Automation, Computers, Electrical and Energy Engineering Department, University of Petrosani, 332006 Petrosani, Romania; 2National Institute for Research and Development in Mine Safety and Protection to Explosion—INSEMEX, 332047 Petrosani, Romania; 3Management and Industrial Engineering Department, University of Petrosani, 332006 Petrosani, Romania

**Keywords:** visible light communication, sensors, methane, pit gas, environmental monitoring, sensors, detection, tracking

## Abstract

The continuous monitoring systems developed for health and safety provisions in underground mines have always been a deep concern in mining activity, since this is one of the most dangerous work activities in the world. All mining activities must follow strict regulations that pose an increased responsibility for the design and implementation of novel technologies aiming to enhance both the workers’ and equipment’s safety in underground mines. Coal mines are some of the most dangerous, with high risk of explosion due to pit gas; therefore, we propose the use of visible light communication for local wireless transmission and optical fiber for remote-cabled transmission. We address, in this article, a complete system comprising real-time personnel tracking and environmental methane measurement, as well as the transmission of data, with a detailed explanation of the complete system and technical description with practical solutions applicable for industrial use.

## 1. Introduction

Disasters in underground mines caused by the ignition of pit gas (a mixture of methane gas with coal dust) followed by explosions produce human victims, as well as significant financial losses for mining companies every year, worldwide. To enhance both the workers’ security and mine’s safety, a reliable, embedded system of real-time environment monitoring and personnel tracking in these harsh and confined environments, as underground coal mines are, must be developed.

The design, implementation and maintenance of solid and reliable data communication systems from the underground spaces of mining companies to the surface are challenging tasks because of the strict regulations regarding the personnel’s health and safety, the equipment’s and machineries’ safety, the harsh environment, confined spaces, difficult preservation and continuous changing of the mine topology, all of which result in high implementation and maintenance costs.

Many solutions based on novel technologies have been gradually adopted for local and remote data communication in underground mines. The lack of GSM signals underground, poor cellular transmission and electromagnetic interferences due to the use of RF waves that affect the operation of electronic devices allow optical transmission technologies to become a reliable candidate for a solid communication system. The optical wireless communication technologies with indoor applications, Visible Light Communication—VLC, Light Fidelity—LiFi, Optical Camera Communication—OCC, although novel, have remarkable benefits over the classical communication scenarios based on RF waves for local, short and medium distance, fast, green, safe and low-cost transmission.

The first part of this article addresses the regulation and special requirements for electrical and optical equipment developed for wireless and cabled communication in underground coal mines. The pit gas in coal mines (where the explosion hazard is very high) imposes very strict regulations that must be followed in order to ensure a high degree of health and safety for both personnel and equipment during working activities in the underground spaces.

An overview of the current systems used worldwide for Personnel Tracking and Monitoring Systems (PT&MSs) with industrial practical applications follows. A short literature review regarding the worldwide research conducted on the use of visible light in mining exploitations is also presented.

The system that we propose for personnel tracking and essential environmental parameters’ continuous monitoring is a special design for coal mines where disasters frequently occur because of the pit gas concentration and its fast ignition. The communication system described here aims to allow data acquisition from underground and transmit them, in real time, to the mine surface, in order to monitor the methane concentrations and supervise the personnel on the main galleries in the coal mines. Data stored in a digital form are used for further analyses and interpretations in order to avoid unwanted events, such as accidents and explosions. We present here a description of the equipment already approved for use in coal mines (ATEX certified [1]) that are considered in the design of the entire remote-cabled data transmission network. Furthermore, we present a detailed analysis of the local transmission and warning system with technical specification of the equipment we designed for a simplex wireless communication based on visible light. The results of the simulation conducted with the use of dedicated applications are discussed and interpreted to fully appreciate both the advantages and drawbacks brought about by the implementation of such a system in coal mines; a system that aims to considerably increase personnel safety and keep the network’s cost as low as possible at the same time.

## 2. Requirements for Electrical and Optical Equipment Developed for Wireless and Cabled Communication System in Underground Coal Mines

### 2.1. Health and Safety Requirements for Underground Coal Mines

The explosive atmosphere in underground coal mines has to be continuously monitored due to the presence of potentially explosive gases, especially methane gas (as well as coal dust). Electrical and electronic devices must be designed so as to avoid the ignition of the explosive atmosphere by using specific and adequate types of protection for use in underground mines (for example intrinsic safety) [2]. All systems in underground firedamp mines must adhere to the applicable legislation and only adequate Ex protected devices (for example “intrinsically safe” devices) must be used. Intrinsically safe devices might also be required in non-coal mines that present explosion hazards.

Equipment and systems in explosion-proof construction used to transmit data locally and remotely in potentially explosive atmospheres can be designed so as to comply with the following types of protections: Intrinsic safety “i”—the specific requirements are mentioned in the standard SR EN 60079-11 [2];Optical radiation “op”—the specific requirements are mentioned in the standard SR EN 60079-28 [3];Along with the specific standards, the requirements of SR EN 60079-0 also apply. This contains the general requirements for electrical equipment designed for use in explosive atmospheres [4].

In order to be able to determine exactly how to design and install a local and remote data transmission system in a potentially explosive atmosphere, the construction of the system and its parameters must be well defined.

The SR EN 60079-11 standard contains the specific construction and test requirements for the intrinsically safe devices intended for use in a potentially explosive atmosphere and for the associated devices (which are important for the safe connection to the intrinsically safe circuits entering a hazardous explosive atmosphere). This type of protection is applicable to electrical equipment in which specific technical measures are applied to electric circuits so as to limit the energy and surface temperature and make them not capable of causing the ignition of the surrounding explosive atmosphere [2].

### 2.2. Requirements for Electrical Equipment and Transmission Systems Using Optical Radiations for Group I with Level of Protection (EPL) Ma and Mb

Optical equipment is increasingly used for communications, surveying, sensing and measurement. In material processing, optical radiation of high irradiance is used. The radiation from optical equipment may pass through an explosive atmosphere, in case the installation of such equipment is made inside or close to a hazardous explosive area [3].

Optical radiation, depending on the characteristics, might be able to ignite a surrounding explosive atmosphere. The presence or absence of an additional absorber, such as particles, significantly influences the ignition [1,4].

Regarding the optical radiation, four possible ignition mechanisms are identified, [3]:Surfaces or particles can heat up due to the absorbed optical radiation, allowing them to attain a temperature that can ignite an explosive atmosphere;If the optical wavelength matches an absorption band (of the flammable gas or vapor) it can lead to thermal ignition of a gas volume;Photo dissociation of oxygen molecules caused by radiation in UV range can lead to photochemical ignition;Plasma and a shock wave (that can act as an ignition source) can occur due to direct laser induced breakdown of the gas or vapor (at the focus point of a powerful beam). Solid materials close to the breakdown point can support these kinds of processes.

Due to the possible ignition mechanisms, the first step is to subject the optical equipment to an ignition hazard assessment considering the following principles [4].

Related to the protection against explosion of equipment using optical radiation, three types of protection can be applied. These types of protection that cover the entire optical system [3] are:Inherently safe optical radiation—“op is”;Protected optical radiation—“op pr”;Optical system with interlock—“op sh”.

In case that the beam strength exceeds a level considered to be inherently safe, and an absorbing solid material exists in the beam, it can cause the occurance of a hot spot and an ignition source (or, in case of pulses, the conditions for a breakdown), as can be seen in Figure 1.

#### 2.2.1. Requirements for Inherently Safe Optical Radiation “op is”

In case of inherently safe optical radiation, the visible or infrared radiation is incapable of providing sufficient energy to ignite an explosive atmosphere under normal or specified fault conditions. This concept is based on the limitation of the optical beam to a safe value (considering that from the identified ignition mechanisms in the visible and infrared spectrum, the ignition of an explosive atmosphere by an optically irradiated target absorber requires the least amount of energy, power or irradiance). This concept applies to unconfined radiation (not requiring the maintenance of an absorber-free environment) [3].

Continuous wave radiation

The values presented in Table 1 and Table 2 related to optical power or optical irradiance must not be exceeded. These values are categorized by equipment group and temperature class [3].

The following options are available (as an alternative to compliance with Table 1) [3]:In case of irradiated surface areas above 400 mm^2^, in order to establish the temperature class, the maximum temperature measured on the irradiated surface shall be used (with no limit on irradiance). Nonhomogeneous optical beams shall be considered when making the temperature measurement.In case of limited irradiated areas lower than 130 mm^2^, the values in Table 2 can be used for maximum radiated power values for temperature classes T1, T2, T3 and T4.The ignition tests in accordance to with SR EN 60079-28 shall be passed [3].

2.Pulsed radiation

In case of pulsed radiation, the optical pulse duration for Ma or Mb equipment must be measured under fault conditions (considering the over-power/energy fault protection criteria according to the inherently safe concept) by using, for example, an electrical oscilloscope to measure the pulse duration of the voltage (at the input to the optical device for each fault condition) [3].

3.Additional requirements for optical pulses for Group I equipment

The output parameters must not exceed 0.1 mJ/mm^2^ for pulse lasers (or pulse light sources) with pulse intervals of at least 5 s in case of equipment incorporating optical sources for EPL Ma or Mb. If pulse intervals are less than 5 s, the radiation source is regarded as a continuous wave source [3].

#### 2.2.2. Requirements for Protected Optical Radiation “op pr”

According to this concept, the radiation is confined inside optical fiber (or another transmission medium), assuming that the radiation cannot escape from the confinement. The performance of the confinement will define, in this case, the safety level of the system, “op pr” [3].

Optical components have to be adequate regarding the ratings and temperature range [3].

The release of optical radiation into the atmosphere is prevented during normal operating conditions by the optical fiber or cable. In case of EPL Mb, fiber cables protected by “op pr” have to be provide an additional armoring, conduit, cable tray or raceway. 

Dedicated couplers or joining kits providing a fixed termination can be used in case of internal or external cables that are terminated/spliced from one fiber (from a cable) to another fiber (in a new cable). The cable connection must provide at least an equivalent mechanical strength to that of the cable in case of external termination/splicing. The instructions shall detail the method to perform field connections [3]. For EPL Mb, the requirements for pluggable connections of SR EN 60079-15 apply to optical fiber or cables connected via internal pluggable factory connections [3].

External optical fiber or cable field connections must be in compliance with the requirements for external plug and socket outlets provided in SR EN 60079-0 for the required EPL [4].

#### 2.2.3. Optical System with Interlock “op sh”

The type of protection “op sh” can be applied also in case of non-inherently safe radiation. Radiation is confined inside an optical fiber (or other transmission medium) assuming that the optical radiation cannot escape from the confinement (under normal operating conditions) [3].

The type of protection “op sh” (depending on the EPL) requires the application of “op pr” principles, and additionally, an additional interlock cutoff must be used, as follows [3]:For level of protection Ma, in case of “op sh” applications, protected fiber optic cable “op pr” for Mb must be used together with a shutdown functional safety system that is based on the ignition delay time of the explosive gas atmosphere.For level of protection Mb, in case of “op sh” applications, protected fiber optic cable “op pr” for Gc/Dc must be used together with a shutdown functional safety system based on eye protection delay times.

In case the protection by the confinement fails, the interlock cut-off must operate in a shorter time than the ignition delay time (or the delay time for eye protection). The interlock cut-off delay time in case of Group I equipment shall be less than the boundary curve of Figure 2 (represented by the curve fit to minimum ignition delays with a safety factor of 2 included).

Aside from the health and safety requirements for underground coal mines, the topology of each mine has a significant impact on the design, installation and use of any system of local and remote communication. Moreover, any communication system must be updated and improved continuously to best fit the needs of the growing mine. As the mine and its operations continue to grow and become more and more advanced, the communication system must grow with it. Along with the mine changes, the underground local and remote communication system changes as well.

## 3. The Current Personnel Tracking and Monitoring Systems with Local Wireless and Remote-Cabled Communication for Health and Safety Provisions

### 3.1. The Underground Monitoring Systems

The Personnel Tracking and Monitoring Systems (PT&MSs) that we propose for underground mines are developed to be easily deployable, scalable, robust, with long life, being able to quickly increase the coverage as the mine extends, while the communication architecture is designed to be resilient to blockages and quickly responsive in case of the nodes’ transmission failures.

Currently, the monitoring systems used in the underground mines during operations perform data acquisition of several parameters from different fixed measurement points, with the support of portable and/or mobile devices. The portable devices need to be continuously worn by the personnel underground and the mobile devices can be moved to different places according to extension of the mining activity.

The large mining companies worldwide have well-defined procedures for the following physical and chemical parameters to be monitored underground on a regular basis: the air and rocks’ temperature, air humidity, differential pressures, atmospheric pressure, direction and velocity of air circulation, oxygen (O_2_) and toxic and hazardous gases with specific concentrations.

Several sensor technologies are being used for monitoring air quality and the environmental ambient conditions, such as [5]: electrochemical sensors, semiconductor sensors, as well as optical and optical fiber sensors. 

The electrochemical sensors are used for monitoring atmospheric gases, such as O_3_, CO, H_2_S, H_2_, NO, NO_2_ and SO_2_ [6,7,8]; the semiconductor sensors are used to monitor the atmospheric gases such as CO, CO_2_, O_3_, ammonia (NH_3_), CH_4_ and NO_2_, as well as ambient temperature, humidity and atmospheric pressure [9,10]; and the optical and optical fiber sensors are used for ambient monitoring of humidity and temperature, as well as for monitoring the atmospheric gases such as SO_2_, NO_2_, O_2_ H_2_, CH_4_ and NH_3_ [11,12,13].

The atmospheric parameters under observation must be within a certain safety range of values allowed by the regional standards in force. Compliance with these conditions ensures part of the health and safety of work in underground mines.

Most of the systems worldwide that aim to monitor, in a continuous manner, the atmosphere in the underground mining environments [14,15,16,17,18] consist of the following main elements: The local Programmable Logic Controllers (PLCs) situated underground that allow the acquisition, control, storage and data visualization of the values related to the concentration of atmospheric parameters and are able to provide various commands and notifications with optical and audio signals as fixed or mobile warning systems;Fixed and mobile elements that measure a number of parameters (sensors/transducers);The power supply network of the equipment within the monitoring system;The transmission network for data communication (wireless or cabled data communication channels, switches, etc.);The PC server, usually situated at the surface of the mine company.

In order to monitor the atmospheric parameters within underground mining operations, the PLCs can be connected in various network topologies (bus, ring, mesh, star, etc.). The choice of network topology is made according to the technical–economic benefits/drawbacks of each topology type.

Discontinuous monitoring can be performed with the support of mobile and portable devices that monitor the atmospheric parameters in underground mines. These portable devices, compulsory to be worn by miners the entire time they are underground, acquire environmental data that are stored in a memory card. 

Data can be downloaded, read and analyzed at the end of each miners’ work schedule. Currently, portable/mobile devices have optical and acoustic warning signals that are activated when the monitored variables exceed the threshold values.

For example, the portable multi-gas detector ALTAIR 4XR allows the transmission via Bluetooth of alarms and data recorded, over a distance of 10 m, to one or more associated receivers (ECOM SMART EX 01 smartphones), which use the free application MSA ALTAIR Connect for Android [19]. The Microtector III G888 portable multi-gas detector is capable of transmitting, by radio frequency (RF), the ID, alarms and data recorded over long distances (868 MHz for EU −700 m, respectively, 915 MHz for US −300 m) to one or more receiving devices [20].

The PMA-2008 is an anemometer that measures the air circulation speed in underground mines. This device does not store or transmit the data measured [21].

Currently, in addition to the monitoring portable devices, companies have developed zonal mobile devices, such as BM25 (Figure 3a), able to monitor, simultaneously, five types of gases. In case of exceeding the threshold values of the monitored gases, the BM25 device sends acoustic and optical signals as a warning emergency/error status. In this case, all the BM25 devices on the same network enter the same state and display the specific error message on the LCD screen (this error message is deleted when the BM25 device that initiated the error returns to the normal operating state). It has an internal memory that allows the storage of data and allows a wireless transmission (over a distance of about 1000 m) via RF (2.4 GHz with a transmission power less than 100 mW, according to IEEE 802.15.4 standard) of waning signal/errors/measured data to one or more reception devices. It can be used both as a standalone device and in a mesh network with maximum 30 devices. An example of a mesh network, consisting of 6 BM25 devices, is shown in Figure 3a [22]. In a certain area there can be up to 16 independent networks without interference.

Within a mesh network, BM25 can connect peer-to-peer with all other units, each node being able to send/receive/retransmit data to any other unit connected in the network. In the case of power supply, this comes from an intrinsically safe source.

When an MX40 control panel is inserted into a mesh network, all BM25 devices in the network can transmit warning signals, errors or measured data to the MX40 PLC (when the devices work in the controller mode).

Each slave device has a certain address that can be set in the range (1–30) as well as a certain network ID which can be set in the range (0–15).

Within a network, all BM25 devices must have the same network ID. An example of a mesh network, made around an MX40 PLC, is shown in Figure 3b [23].

The MX40 PLC can wirelessly control BM25 slave devices; therefore, in case of exceeding a threshold value of an atmospheric parameter, the PLC can transmit wirelessly, to the other BM25 devices in the network, the commands necessary to activate the emergency signals. It has a memory card that allows data storage in *.csv format (min/max, averages, alarm event states, faults and calibration).

Data can be viewed on the PLC’s Human Machine Interface (HMI) display or downloaded to a PC using the Log File Viewer program.

On the other hand, based on the input adapter modules, the MX40 PLC can also acquire data from fixed measuring elements (which have current outputs: 4–20 mA or digital outputs, based on RS485 and the MODBUS protocol). In the case of a mixed network, the MX40 PLC can monitor a maximum of 32 measurement channels. The MX40 PLC has a standard RS485 output, which allows communication via the MODBUS protocol within a distributed monitoring and control system (SCADA). 

Using the Sitewatch option, the MX40 PLC can also communicate via Ethernet (e-mail) or GSM (SMS) technology. In case of alarms or malfunctions, e-mails/SMS are forwarded together with data acquired by MX40. Within a network, the MX40 PLC can communicate with other compatible devices. However, Teledyne Gas & Flame Detection does not produce any portable devices (such as those shown in Figure 3) that can communicate with the MX40 PLC [23].

The company AMS Equipment Limited produces a multi-gas-zonal mobile device (similar to BM25), called Radius BZ1 (detects seven gases), which allows communication via RF (2.4 GHz) with mobile-personal multi-gas devices, from the series Ventis PRO (like those shown in Figure 3). Devices that communicate with BZ1 are included in a mesh network (maximum 25 devices in a network; in a certain area there can be up to 10 independent networks).

Within a mesh network can be used: BZ1 devices, Ventis PRO personal gas devices, as well as RGX Gateway and TGX Gateway devices covering a maximum area of 300 m. However, the Radius BZ1 device is not certified for use in underground mines [24].

### 3.2. The Underground Personnel Tracking and Remote Communication Systems

The tracking systems aim to gain, continuously, in real time, information regarding the workers’ position in the underground mine, and keep, in the server located on the mine’s surface, an up-to-date database of the personnel’s presence and location underground for a certain period of time.

Data gained locally in underground spaces are remotely transmitted to the surface of the mine in order to be continuously observed and evaluated in real time with the aim of avoiding any unwanted events, such as accidents and/or explosions.

Many local and remote communication solutions are now used worldwide for mining companies for underground operations, such as through-the-earth (TTE) transmission performed by magnetic induction, with frequencies below 30 kHz [25], leaky feeder systems (LFS) based on UHF and VHF [26,27], Wi-Fi [28,29], cellular networks with Long Term Evolution (LTE) network technology, or radio systems for short range communication Ultra-Wideband (UWB), Voice over Internet Protocol (VoIP) [30].

The majority of underground mines today use the two-way land mobile radios systems with LFS to address their voice communication needs, while wireless data transmission relies on Wi-Fi networks.

A wireless underground tracking and monitoring system based on mobile sensors and access points embedded into the communication backbone of the mine can provide reliable and real-time monitoring capability even after a mine disaster happens, which can greatly enhance both mine safety and the company’s productivity [31].

The most adequate local wireless and remote-cabled data communication system must be selected according to a balanced analysis between several technical criteria and economic benefits and/or drawbacks of each type of communication, taking also into consideration any restriction regarding the use of certain technologies in hazardous environments.

Local and remote transmission systems are either cabled (copper wire or optical fiber) or wireless communication (using the radio and optical part of the electromagnetic spectrum) [32,33,34,35] as in Figure 4:

Local cabled communication (LCC) for short link between the measuring modules (MM) and PLC using RS485 serial connection, Profibus (Process Field Bus), Modbus protocol, Canbus (Controller Area Network bus), AS-Interface (Actuator Sensor Interface, ASi), Hart communication protocol (Highway Addressable Remote Transducer), etc;Local wireless communication (LWC) for short link between MM and PLC using Global System Mobile (GSM), Wi-Fi, Bluetooth, Bluetooth Low Energy (BLE), world interoperability for microwave access (WiMAX), ZigBee, Z-Wave, 6 LowPAN, RFID, Ultra-Wideband (UWB), Near Field Communication (NFC), Light Fidelity (Li-Fi), Visible Light Communication (VLC), Optical Camera Communication (OCC);Remote cabled communication (RCC) for extended link between PLCs and switches (SWs) using Ethernet technology with cooper cable (coaxial or shielded twisted pairs—STP), Power Line Communication (PLC), Power over Ethernet (PoE), RS485, Profibus, Modbus, Canbus, AS-Interface, Hart, etc.;Remote wireless communication (RWC) for extended link between PLCs and SWs using GSM, Wi-Fi or Li-Fi;RCC for extended link between SWs and PC server using Ethernet technology with cooper cable (coaxial or shielded twisted pairs—STP), Power line communication (PLC), Power over Ethernet (PoE), RS485, Profibus, Modbus, Canbus, AS-Interface, Hart, etc.;RWC for extended link between the SWs and the PC server using: GSM, Wi-Fi.

Using Wi-Fi technology, SCADA and Ethernet networks, a near and far field proximity detection and collision avoidance system for underground vehicle and personnel tracking with remote evacuation signaling was developed and applied by Becker company for Mining in South Africa with the aim to increase health and security in underground mines [36].

However, there are number of drawbacks when RF-based wireless communication technologies are used in underground mines: the short range of the RF technology (that operates in the 2.4 GHz band) cannot provide large coverage in underground environments [37]; is not a proper solution in some underground mine environments where RF signals are limited by strict rules, or their use is even forbidden, as is the case of coal mines with pit gas; technologies such as RFID, Wi-Fi or ZigBee involve additional costs for development and dedicated configurations; low precision that is typically between a few tens and several hundreds of meters.

Therefore, to avoid potential hazards and accidents in underground mines, alternative solutions have to be developed and applied to reduce the downsides brought about by the use of RF signals. The last decade’s technological progress in OWC technology find its place in underground mines with its novel applications, especially as a fast and reliable system that supports the workers’ safety and security during mining activities.

### 3.3. The PT&MS with VLC Technology for Underground Mines

Substantial worldwide efforts and many research projects have been dedicated to identifying the key risk factors that lead to accidents [38] with human casualties and substantial costs in underground mines. Regardless of the underground exploitation, the most frequent accidents and hazards are related to toxic gases, pit gas in coal mines, inappropriate lighting, rocks or roof falling, narrow spaces and irregular floor or low-quality ventilation. Poor air quality comes from high level of dust with different shapes and dimensions of suspended particles in air, due to exploitation itself or from constant or instant emission of gases specific to mining activity [39]. The most viable solutions developed so far, aiming to prevent or avoid accidents, have been applied in some mining exploitations worldwide [40,41,42]. 

Optical communication uses parts of the electromagnetic spectrum (visible light, infrared and ultraviolet) for its novel developed technologies and applications (Infrared IR, VLC, LiFi, OCC, Free Space Optics—FSO) aiming to improve the local and remote transmission capability and become an energy efficient support system for a faster and greener ubiquitous transmission network [43].

VLC technology is appropriate to replace or work as a complementary technology in underground coal mine local communication networks due to some benefits that optical wireless transmission has over the RF communication:LED light is the most suitable type of light to be used in mines for lighting, such as in miners’ cap lamps but also in the luminaires [44];LEDs, as lighting sources used underground, are already set into the illumination system on the main galleries underground;The key characteristics of LEDs have been significantly enhanced to be able to convey data at the same time with the illumination;Industrial LEDs are also engineered to operate in extremely harsh environments, i.e., they are explosion-proof, resilient to shockwave, immunity to vibration.

Personnel positioning underground and/or equipment’s tracking solutions developed with wireless communication technologies based on RF, IR or VL in spectrum range use Received Signal Strength (RSS), Time of Arrival (ToA), Time Difference of Arrival (TDoA) or Angle of Arrival (AoA) techniques [45].

The RSS technique requires a direct Line of Sight (LoS) topology and a strong optical link between the optical transmitter (oTx) and receiver (oRx) to work properly. The ToA technique uses the necessary time of the optical signal to travel from oTx to oRx. The oRx’s clock has to be synchronized with the oTx’s, although the synchronization in VLC systems is a difficult task. The TDoA uses Time Division Multiplexing (TDM) or Frequency Division Multiplexing (FDM) techniques in an LoS topology. At least three sources have to be sensed by the targeted entity (person or equipment) to be localized. The AoA uses also an LoS topology and both the incidence and irradiance angles are measured. Other sensors (accelerometers, for example [46]) can be used to compensate the effect of tilt oRxs in combination with a multiple photodetector (PD) array [47].

The optical signal propagation in underground spaces undergoes many physical phenomena (such as reflection, absorption, diffraction and scattering) because of the complex environment (ground, walls and ceiling) in the main galleries underground and the workface; therefore, accurate data regarding the channel Impulse Response (CIR), Root Mean Square Delay Spread (RMS DS), optical Path Loss (oPL), Signal to Noise Ratio (SNR) and Bit Error Ratio (BER) have to be carefully considered in order to achieve a high-quality transmission of the optical signal [48]. The mining activity itself produces dust particles that have a high negative impact on both accuracy and the length of the optical link in VLC networks [49].

Several monitoring and tracking solutions have been investigated so far for various mining exploitations with many applications [50,51,52,53,54,55].

The use of visible light communication in underground mines is also discussed by a number of authors worldwide. For example, in the paper “Communication system for underground mines using Li-Fi 5G technology” the authors [56] present an overview of a Li-Fi 5G communication system that includes a light/voice signal for passing emergency information to the worker under risk conditions.

A location system for an underground mining environment using visible light communications was proposed by Iturralde et al. [57] in 2014 and an alarming system for an underground mining environment using visible light communications was presented by Farahneh et al. in 2017 [58]. A hemi-dodecahedron angle diversity receiver in conjunction with the maximum ratio combining scheme both in mining roadway and mine working face for a realistic VLC system is proposed by Jativa et al. [59]. Firoozabadi et al. [60] presented a novel frequency domain visible light communication, a three-dimensional trilateration system for localization in underground mining. Seguel et al. [61] made an extensive presentation on the potential and challenges of VLC-based indoor positioning systems in underground mines in 2017. A robust localization system for underground mines using VLC technology has been proposed and analyzed in detail by the author Seguel [62]. The possibility of using VLC with Augmented Reality (AR) technology for an enhanced environmental monitoring, tracking and localization system in the coal mine was also proposed [63]. A similar concept developed for underground positioning and monitoring system is presented in [64]. The concept of communicating the ID’s lamp by its LED’s light to the Access Points embedded into the illumination network in [64] is developed around Arduino PCB and the prototype is tested in the laboratory to prove the concept. However, in this article, the entire electronic and optical system is enhanced and developed with parts according to the highest demanding requirements for any equipment that has to be used in an environment with pit gas. The equipment developed and presented here aim to pass all the tests conducted for an ATEX certification. Moreover, beside the lamp’s ID, the values of the methane gas acquired from the underground environment are also sent by the miner’s cap lamp, and the optical transmission is also assisted by Bluetooth technology when the LOS optical link is blocked.

## 4. The Proposed Underground PT&MS with Hybrid (RF/VLC) Local-Wireless Communication

As the literature in this area shows, there is a trend to use underground tracking and monitoring systems that locally allow data to be obtained and sent, in real time, to the mine surface, resulting in an advanced system of sharing in a global network that spreads over all the underground mine’s spaces.

Following the detailed analysis already presented, several research directions have been identified that need to be developed in the future. These research directions aim at introducing new network topologies between fixed, mobile and portable devices, as well as the possibility of using the wireless communication technologies (GSM, Wi-Fi, ZigBee, Bluetooth, BLE, VLC, OCC, Li-Fi, etc.) between these devices. All the previously mentioned research directions can be corroborated with the progress made in the field of automatic timekeeping at work, as well as with the progress in the field of automatic positioning of personnel within the underground mine.

In this article, we describe detailed research on the possibility of using the VLC technology in a hybrid system (VLC/RF, local wireless/remote-cabled communication technologies) that aims to continuously monitor both the atmospheric parameters in an underground mine and track the miner’s position on the main galleries. The tracking system is designed not only to localize each miner’s position on the main galleries underground, but to keep an up-to-date situation of the personnel’s position, identity and the values of methane gas acquired in certain areas underground.

This research uses the progress obtained in the field of automatic positioning and timing of personnel in underground mines, based on VLC technology. The architecture of the system that we propose aims to monitor the environmental parameters, track the miners and communicate, in real time, the values of obtained data to the surface of the mine. The network of the PT&MS has a star topology, as presented in Figure 5.

In Figure 5, the following notations are used:

$HMI_{i}$; $i=\bar{1,n}$ Human Machine Interfaces; ADP—adapter module used in bidirectional RS485/fiber optic conversion;PLC—p Programmable Logic Controllers;$T_{i}$; $i=\bar{1,n}$ transducers used for measuring the physical and chemical parameters of the atmosphere in underground mining; $ME_{i}$; $i=\bar{1,n}$ elements for measuring the physical and chemical parameters of the atmosphere within underground mining operations; $EE_{j}$; $j=\bar{1,m}$ a number of execution elements (electric cars, power supply sources, etc.); $B_{i}$; $i=\bar{1,n}$ audio signal warning device; $L_{i}$; $i=\bar{1,n}$ optical signal warning device; $A_{i}$; $i=\bar{1,n}$ communication antennas; IB—intrinsic safety barrier;PC—personal computer; SWITCH—switch for optical fiber.

Each PLC in the monitoring system component allows the acquisition/control/storage and local visualization of data. The data acquired by the PLCs are transmitted to the surface to the process computer (server) using a fiber optic network and a switch. The configuration of the PLCs can be performed both from the dispatcher and from the corresponding HMI keyboard. The PLCs are queried by the server at time intervals that can be set from the dispatcher (the PLC query period is in the range: 0.3 s–4 min).

Each measuring element (placed in a certain fixed place within the mine) has a certain transducer and an HMI used in visualizing the value of the parameter (physical/chemical) measured. In case of exceeding the safety range of the parameter, the device emits acoustic and visual warning signals through the speaker and the warning lamp. The PLCs interrogate the measuring elements at short time intervals (ms/µs), being able to control various execution elements in order to control some atmospheric parameters, and, in case of exceeding the safety range (s) of the monitored atmospheric parameter (s), they decouple the electricity and perform the acoustic and visual warning of the workers in the respective workspace.

The server allows the storage of acquired data over long periods of time and can allow the analysis and predictive management of the mining security system. Through PCs and monitors, the purchased data can be viewed by the dispatcher. In case of exceeding the safety range(s) of the monitored atmospheric parameter(s), the server emits optical and acoustic warning signals in the dispatcher’s room(s). In case of explosions, the server emits acoustic and optical signals to the entire mining unit, being able to automatically send SMS messages and predefined voice messages (based on a GSM modem) to certain telephone numbers. The messages can be sent to the telephone numbers allocated to the ambulance services of the medical units or institutions located in the immediate vicinity of the mining unit, as well as to the director, sector heads and mining rescuers, etc. 

The data transmission between the components of the monitoring system is performed using guided physical communication channels. In Figure 5, the measuring elements are interrogated by the PLC by means of ADC (Analog–Digital Converter) adapter modules using one of the standard signals: in voltage: 0–10 Vdc; current: 4–20 mA.

### 4.1. Description of the Embedded PT&MS with Local Wireless Hybrid Communication

#### 4.1.1. The Miner’s Cap Lamp as Optical Transmitter (oTx)

The hybrid system that we propose is based on a fixed monitoring system, similar to that in Figure 5, where zonal devices and multi-gas portable devices are used in addition to other fixed elements for measuring atmospheric parameters. They communicate in a local wireless network via VLC technology, as well as Bluetooth.

A similar product with fewer functions than the one we propose has already been developed and used underground, called the mining cap lamp IYONI II, developed by GfG (as seen in Figure 6) [65].

The miner’s cap lamp IYONI II [66] with a 3 W halogen bulb fulfills two functions: lighting the workspace; and acts as portable gas detector with a continuous monitoring function and a warning signal (acoustic and optic), in case of exceeding the threshold value of the gas. These two functions are activated by a switch. It is available with various gas sensors: CH_4_ or CO two-gas sensor, or CH_4_ and CO single-gas sensor. In case of the highest current consumption when CH_4_ and CO sensor are used, the consumed current of the whole assembly does not exceed 970 mA. If the gas concentration exceeds a predetermined threshold value, the optical alarm (flashing) is activated and an LED indicates which gas is in a dangerous concentration. On the other hand, an additional acoustic warning signal (75 dBA) is activated.

We propose a new type of miner’s cap lamp that fulfills the same functions of the IYONI II device and additionally has the VLC module embedded and stores the measured data on a memory card. Therefore, data acquired by the miner’s cap lamp can be wirelessly communicated to a number of access points embedded into the remote cabled communication network that is fixed on the gallery’s ceiling in the underground mine (see Figure 7a).

The wireless local transmission between the miner’s cap lamp and the access points is based on visible light communication and Bluetooth.

Data stored on the card can be downloaded in a docking station (on the surface of the mine), which allows both the calibration of the device and the charging of the battery embedded into the miner’s cap lamp.

The miner’s cap lamp that we propose (Figure 7b) allows an automatic communication of data measured (instantaneous gas concentrations) and the unique ID of the lamp (automatic timekeeping) via VLC technology.

The unique ID of the lamp is automatically connected with the worker’s identity since the portable miner’s cap lamp is worn by the same worker the entire time it is underground. These data are wirelessly communicated in a Line of Sight (LoS) topology to VLC optical receivers (oRx) that act as access points of this wireless optical network.

This new miner’s cap lamp that we propose also has a Bluetooth transmission/reception device (a u-Blox ODIN module—W262 certified ATEX). In this way, the miner’s cap lamp becomes a duplex communication wireless network with transmission (VLC/Bluetooth) and reception (Bluetooth).

Each oRx with the u-Blox ODIN—W262 Bluetooth Tx/Rx module also has an acoustic and optical warning system, which is activated when the threshold value of the monitored atomic parameter within the mine is exceeded. Thus, all the oRx become a complete transmission (RS232/Bluetooth)/reception (VLC/RS232) system [67].

The VLC wireless transmission also allows communication of the worker’s position underground since each fix oRx has also its own ID. All oRx are connected via RS485 to a local PLC, which also includes an HMI. The HMI allows a local viewing of measured data. In case of exceeding the threshold value of the monitored gas, the miner’s cap lamp designed by us emits both acoustic and optical warning signals, similar to IYONI II.

In this case, the miner’s cap lamp transmits data, either via VLC (if the lamp has a viable communication channel) or via Bluetooth. Immediately after receiving the gas concentration outside the safety range, the VLC transmitting/receiving station-fixed activates its own acoustic and optical warning system and then transmits the information to all devices in the network. Therefore, all devices (PLC, VLC transmitting/receiving stations-fixed, fixed measuring elements) that receive this information via RS485 activate their own acoustic and optical warning systems. Moreover, the local PLC transmits commands to disconnect the electricity from the execution elements located in the risk area.

The electrical diagram of the proposed miner’s cap lamp presented in Figure 8. This wiring diagram is based on some of the electrical/mechanical components of the GEN4—NIOSH miner’s cap lamp [68].

The GEN4 miners’ cap lamp consists of:An 8-cell Li-ion battery (output voltage: (3.5–4.2) Vdc; capacity: 17.6 Ah);Three LEDs;An LED driver consisting of LTC3220 (driver with 18 independent sources of constant current 20 mA (total output current: 360 mA) controlled by I2C [69];The upper part of the battery and electric cable, as well as a part of the mechanical and optical components of the headlight component.

The layout of the three LEDs in the headlamp lights GEN4-NIOSH is presented in Figure 9. This lamp is developed around a gas sensor (from SGX Sensortech) and the ATmega328PB AVR microcontroller (3.3 Vdc/8 MHz) [70].

The main photometric characteristics of the miner’s cap lamp are presented below:LED3 is a XPGWHT-L1-0000-00H51 (white color, standard CRI) produced by Cree; it has a maximum consumption of 80 mA (see Figure 8) with a luminous flux of 139l m/350 mA, is placed at an angle of 12° on the mounting plate (see Figure 9) [71]; it has an elliptical beam lens (FCP-E1-XPE1-HRF) and is intended for the vertical lighting of the mining workspace [72];LED2 is identical to LED3, with a maximum consumption of 160 mA (see Figure 8). This LED is placed in a holder with an elliptical beam lens (FCP-E1-XPE1-HRF), at an angle of 0° on the mounting plate. LED2 is intended for horizontal lighting of the mining workspace;LED1 is identical to LED3 and LED2, with a maximum consumption of 120 mA (see Figure 8). This LED is placed in a holder that has a narrow beam lens (FCP-N1-XPE1-HRF), spot beam, at an angle of 0° on the mounting plate. LED1 is intended for data transmission based on VLC technology.

In order to transmit data through VLC technology, an N-type MOSFET transistor (Q1—see Figure 8) is used, using variable Pulse Position Modulation (VPPM) (according to the IEEE 802.15.17 standard) [73]. In order to implement the VPPM modulation, the Pulse Width Modulation (PWM) pin (PD5) of the ATmega328PB microcontroller is used. The electrical diagram in Figure 8 allows the data transmission also through the On–Off Keying (OOK) modulation. 

OOK is a fair trade-off between performance and complexity, considering that off-the-shelf hardware can be easily implemented with this modulation technique. Different, more complicated schemes consider different durations in order to transmit additional data. This technique is an analog to unipolar encoding. Although it is easy to implement, there are several issues regarding illumination control and data throughput. 

In order to reduce the light interference on the VLC-fixed receiver, data is transmitted with a single LED (LED1). At the time of data transmission, the other LEDs, LED2 and LED3, are off. The N-type MOSFET used in Figure 8 is MCP87130.

The miner’s cap lamp that we propose is developed around the 8-bit microcontroller, ATmega328PB, which has an 8MHz external quartz resonator (ECS-80-18-5G3X-JGN-TR) with a load capacity of C_7 = 18pF [74]. The microcontroller, as well as the rest of the devices in the miner’s cap lamp component, are powered by a 3.3 Vdc DC voltage source, made around the 8-cell Li-ion battery and the MIC29300-3.3WU voltage regulator (see Figure 8) [75]. At the input and output of the voltage regulator MIC29300-3.3 WU, two capacitors are used, which have a capacity of C_5 = 10 μF. At the output of the 3.3 Vdc DC voltage source, an intrinsically safe barrier is used, made of an EMI filter, three Zener diodes and a fuse.

The Bluetooth module used in the miner’s cap lamp is u-Blox ODIN-W262. This module is a multi-radio part that allows both Bluetooth communication (dual-mode: BR/EDR v2.1 or BLE v4.0) and Wi-Fi (dual-band: 2.4 GHz and 5 GHz, respectively). ODIN-W262 allows point-to-point communications and point-to-multipoint communications and can have simultaneous Wi-Fi and Bluetooth connections. The Wi-Fi communication of the ODIN-W262 module complies with the IEEE 802.11 a/b/g/n standard. Wi-Fi transmission/reception can be performed over a maximum distance of 250 m, while Bluetooth transmission/reception (Tx/Rx) is less than 100 m. In the case of Bluetooth communication using Bluetooth Basic Rate/Enhanced Data Rate (BR/EDR) technology, the miner’s cap lamp can be connected with a maximum of seven active devices and in the case of Wi-Fi communication, the miner’s cap lamp can be connected to a maximum of ten stations.

The u-Blox ODIN-W262 module is made around an ARM Cortex-M4 processor, which has a 24 MHz quartz crystal and can communicate through UART, SPI, I2C and RMII. The operating voltage of the digital Inputs/Outputs (I/O) of the ODIN-W262 module is 1.8 V. The ODIN-W262 module has three internal DC/DC converters, which generate the following voltages: 1.1 Vdc; 1.8 Vdc/100 mA (voltage is available between pin A2 and GND-see Figure 8) and 2.7 Vdc.

In order to communicate via UART between the ODIN-W262 module and the ATmega328PB microcontroller, the 4-bit SN74AXC4T245 transceiver is used. The operating voltage of port A of the transceiver is set at 3.3 Vdc via terminal 1 (VCCA), and the operating voltage of port B is set at 1.8 Vdc via terminal 16 (VCCB). The direction of data communication within the transceiver (from port A to B/respectively from port B to A) is established by the control pins DIR1, respectively, DIR2 [76]. Thus, if DIRx (x = 1 or 2) is logical 1, the data transmission is made from the Ax port to the Bx port and if DIRx is logical 0, the data transmission is made from the Bx port to the Ax port as long as the pins with denied logic xOE are 0 logical. When the xOE output activation pins are 1 logic, the Ax and Bx ports are in a high impedance (TSL) state. Pin control with xOE negated logic is performed using the ATmega328PB microcontroller, using two digital pins (PD4 and PD7). 

The data transmission/reception using the ODIN-W262 module is performed using the USART 0 port (pins PD0 and PD1) of the ATmega328PB microcontroller. In order to reset the ODIN-W262 module, the button B3 is used, which is connected to pin A1, as in Figure 8.

In order to measure the CH_4_ concentration, the digital transducer with infrared (IR) technology, INIR-ME 5%, from Amphenol SGX Sensortech, which is developed around an ARM7TDMI microcontroller, is used. The measured value of the CH_4_ concentration is set at the output of the transducer both in digital format (using the serial port of the ARM7TDMI microcontroller in the format: 8 data bits, one STOP and without parity) and analog (using a DAC of the ARM7TDMI microcontroller, which has a resolution of 12 bits). In the case of the miner’s cap lamp, the serial port of the INIR-ME5% transducer is used. This transducer allows the measurement of CH_4_ in the triple range (LOW 0% v.v.–1% v.v.; MID 1% v.v.–4% v.v.; HIGH 4% v.v.–5% v.v.) with automatic switching between intervals, with a resolution of 10ppm. The INIR-ME5% transducer is calibrated by the factory. The CH_4_ concentration measured with the INIR-ME5% transducer is taken over in the ATmega328PB microcontroller, using the USART 1 port (pins PB3 and PB4). The heating time of the sensor in the INIR-ME5% transducer component is 45 s after each start. The maximum response time of the transducer (without dust filter) is T_90_ = 30 s. In the miner’s cap lamp, the interrogation time of the INIR-ME5% transducer is performed at intervals of 4 min [77].

In order to determine the time at which the measurements were made, the miner’s cap lamp uses the real-time clock/calendar circuit, RV-8803-C7 [78]. The RTC circuit has an uncompensated 32,768 kHz crystal depending on the temperature. Instead, the 1Hz frequency, the clock/calendar and the frequencies within the range (64–4096) Hz are digitally compensated by means of an individual circuit, depending on the temperature (the compensation values are entered in an EEPROM, directly from factory), thus obtaining a time measurement accuracy of ±3.0 ppm in the temperature range (−40… + 85) °C, with a current consumption of 240 nA, obtained at a supply voltage of 3.3 Vdc. On the other hand, through the OFFSET value, digital compensation of aging can be performed. The RV-8803-C7 circuit has an I2C communication interface that allows communication with other devices (SCL-pin 8; SDA-pin 1). Moreover, the RV-8803-C7 also has an I/O control port that allows:Alarm interrupts/periodic interruptions of the timer/interruptions related to the periodic updating of the time, which are obtained following some settings that can be made on days/dates/h/min (output with negated logic INT-pin 6);Input for an external event (EVI-pin 7), with interrupt function;Programmable clock output (CLKOUT-pin 2) for peripheral devices (32,768 kHz, 1024 Hz, 1 Hz). This function can be activated/deactivated via the CLKOE input pin (pin 4).

In the miner’s cap lamp, the CLKOUT output is deactivated by connecting the CLKOE pin to GND using a resistance *R_2_* = 100 kΩ.

In addition to those mentioned above, the miner’s cap lamp allows acoustic and optical signaling of the worker at the end of the work schedule. This warning signal can be deactivated via button B2 (connected to the EVI pin via a resistor R2). 

Constantly, at a time interval of 5 min, (through the RV-8803-C7 circuit and the buzzer in the miner’s cap lamp), the worker receives a very short acoustic warning signal, as a short notice that the CH_4_ concentration is being transmitted via VLC to the oRx. The measurement was initiated one minute before this short acoustic warning signal. This allows the optimal positioning of the worker close to an oRx VLC. In case that the transmission via VLC does not take place in due time, (about 30 s) the gas concentration is automatically transmitted to the Access Point via Bluetooth. Each transmission is made at intervals of 5 min and 30 s. 

In order to keep the settings while the lamp battery is low, the RV-8803-C7 uses an external 3 V battery (CR1225). To avoid exceeding the maximum voltage recommended as necessary to supply the RV-8803-C7 circuit, two Schottky diodes (BAS70-05) are used, connected as in Figure 8 [79]. Near the RV-8803-C7 circuit, a decoupling capacitor is used (located between the VDD and GND pins) with a capacity *C*_4_ of 10 nF. 

Within the miner’s cap lamp, the ATmega328PB microcontroller communicates with the RV-8803-C7 circuit through the I2C interface (PE0 and PE1 pins), as well as the PD3 digital pin.

The miner’s cap lamp has an SD card as a reading/writing module, connected to the ATmega328PB microcontroller through the SPI1 interface (pins PC0, PE3, PC1 and PE2 as in Figure 8) with the aim to store the measured concentrations of CH_4_. On the SD card are stored both the concentrations measured by CH_4_, as well as information on the times when the measurements were taken (from RV-8803-C7). In case of exceeding the threshold value of the CH_4_ concentration, the miner’s cap lamp transmits acoustic and optical warning signals through a piezo buzzer and 2 red LEDs that are connected as in Figure 8. In the control of the buzzer (PS1740P02E), a PNP transistor S85550 is used, marked as Q2 in Figure 8, which is controlled with the support of the PWM-PB1 pin of the ATmega328PB microcontroller. The PS1740P02E buzzer produces a unique tone, which has a sound pressure level (SPL) of 75 dB (A)/10 cm when on the PB1 pin PWM pulses of 4 kHz frequency are generated. Within the acoustic and optical signaling circuit, three identical resistors are ($R_{1}=1\;kΩ$). On the other hand, in case of exceeding the threshold value of the CH_4_ concentration, through the I2C interface and the LTC3220 driver, the LED 3 in the miner’s cap lamp will start blinking at intervals of 0.625 s. When data are transmitted via VLC, the LED2 and LED 3 of the miner’s cap lamp stops for 10 s.

The miner’s cap lamp is switched on/off via the push button B1 (see Figure 8). Using the push button B1, the following modes of operation of the miner’s cap lamp can be selected:Low intensity lighting mode. B1 button is pressed once for a few seconds (2–4 s). In this mode, the lighting is provided by LED3;Medium intensity lighting mode. B1 is pressed twice in a row. In this mode, the lighting is provided by means of LED2 and LED3;High intensity lighting mode. B1 is pressed three times in a row. In this way, the lighting is provided by the 3 LEDs in the component of the miner’s cap lamp (LED1, LED2 and LED3);The lighting is switched off by pressing the B1 button four times.

At the time of VLC transmission, LEDs LED2 and LED3 are turned off and the transmission is performed by LED1. After the data transmission, the headlight of the miner’s cap lamp returns to the initial lighting mode set by the worker.

The passive elements used in the circuit (Figure 8) have the following values:$$\;R_{1}=1\;kΩ;\;\;R_{2}=100\;kΩ;\;\;R_{3}=50\;kΩ;\;\;R_{4}=3\;kΩ;\;\;L_{1}=10\;\mathrm{nH};\;\;L_{2}=10\;\mu H\;;\;C_{1}=C_{2}=2.2\;\mu F;\;\;C_{3}=C_{6}=0.1\;\mu F;\;\;C_{4}=10\;\mathrm{nF};\;\;C_{5}=10\;\mu F;\;\;C_{7}=18\;\mathrm{pF}.$$

In Figure 8, the inductor *L*_1_ is made through a 1cm wiring. When the voltage is less than 3 Vdc, in order to prevent a complete discharge of the battery, the miner’s cap lamp has a shutdown circuit.

In the electrical diagram in Figure 8, the CH_4_ transducer can be replaced with other transducers, such as INIR-CD 5% (for CO_2_ monitoring in the LOW range 0% v.v.–1% v.v.; MID 1% v.v.–4% v.v.; HIGH 4% v.v.–5% v.v.) or INIR-ME 100% (for monitoring CH_4_ in the range LOW 0% v.v.–1% v.v.; MID 1% v.v.–4% v.v.; HIGH 4% v.v.–100% v.v.). The electrical diagram in Figure 8 can be extended with other transducers with analog or digital output, which are taken over on the free ports of the ATmega328PB microcontroller.

When the data transmission cannot be performed, either through VLC or through Bluetooth, the related data will be saved on the SD card.

The entire assembly in Figure 8 is inserted into a special box with the IP67 sealing class and is attached to the headlight of the miner’s cap lamp. Regardless of the transducer used, a charcoal dust filter is mounted on the special box of the device near the hole where the transducer is placed.

The miner’s cap lamps, as part of the hybrid system for personnel tracking and monitoring the atmospheric parameters of underground mining operations, can be used with various network topologies.

#### 4.1.2. The Access Points as VLC Optical Receiver (oRx)

The oRx configuration is the most challenging task in the VLC setup. Light emitted by the LED is concentrated using optical elements and filters before hitting the active area of the PD’s surface.

To achieve a high data rate, the bandwidth-limiting effect generated by the yellow layer of phosphor has to be avoided. In order to avoid it, there are some techniques used today in a VLC system: the use of complex modulation schemes involving multiple bits carried by each symbol transmitted (this method merges multilevel modulation techniques like QAM with optical OFDM or DMT modulation). When used with blue filtering, the transmission rate can be extended to hundreds of Mbps (blue filtering in front of PD cleans out the yellow components that have a slow response) [80,81] pre-equalization at the oTx and post equalization at the oRx [82].

The choice of the best suited PD for a specific VLC setup refers to its active area, capacitance and the spectral response, as well as any transit-time limited bandwidth effects [83].

The overall performance of the oRx front end device depends on the bandwidth, sensitivity and active surface of the photodetector (PD), as well as the quality of the communication channel.

In order to achieve the expected data rate, the PD must have: high quantum efficiency, high responsivity, high photosensitivity within its operational range of wavelengths, low noise level, long operational lifetime and minimum response at a wide range of temperature fluctuations.

The circuit of the oRx (Figure 10) must gain a high accuracy of data transmitted at the targeted distance with the best possible bit error ratio (BER).

Designing an appropriate Trans Impedance Amplifier (TIA) for a robust oRx in a VLC setup is not an easy task, since is difficult to establish the correct balance, a proper trade-off between gain and bandwidth. Low input TIAs are appropriate for high bandwidth and low noise performance in VLC oRx setups with low sensitivity drawback. In contrast, high input TIAs are highly sensitive with low frequency performance. Feedback TIAs overcome the above-mentioned drawbacks of the open loop TIAs with small input impedance with high bandwidth and high gains with low sensitivity.

Low input TIAs are appropriate for high bandwidth and low noise performance in VLC oRx setups with low sensitivity drawback. In contrast, high input TIAs are highly sensitive with low frequency performance.

The block diagram of the oRx VLC is shown in Figure 10.

The data sent by oTx (the miner’s cap lamp) through the LED’s light hit the active area of a PD type BPX61 [84] that has a flat-convex lens type LA1951-A [85]. The oRx front end is developed around the ATmega328PB microcontroller. The acoustic and optical warning system in Figure 10 is identical to the one in the miner’s cap lamp component (see Figure 8). The ODIN-W262 Bluetooth module is connected through the SN74AXC4T245 transceiver to the ATmega328PB microcontroller, as in the electrical diagram of the miner’s cap lamp from Figure 8. The RC circuit is used to filter the +5 Vdc voltage (obtained from a DC/DC buck converter developed around the LT1376HV circuit [86]) of the BPX61 PD.

Communication between oRx and DKDP PLC is performed using RS485 (MODBUS RTU protocol), therefore, a UART/RS485 converter is used, made around the ISL32452 circuit [87].

Each VLC reception module has a unique address that is set via six jumpers (a total of 64 addresses). The distance between the oRx modules is set by means of two buttons (BI-for incrementing, respectively, BD-for decrementing). The distance value is displayed on a 7-segment display with 3 digits, which communicates with the ATmega328PB microcontroller using the I2C interface.

The electronic scheme underlying data reception through VLC technology is based on the electronic scheme proposed by Stijn Wielandt [88].

The output current of the PD is transformed into a voltage within the range (0.1–0.4) Vdc by a TIA [88], using the OPA2846 amplifier [89]. The cut off frequency used in the TIA filter is 300 Hz (considering that the human eye cannot detect the LED’s flickering with a frequency higher than 200–290 Hz).

To reduce the direct current of the PD, which occurs due to the luminaires in the underground mine, in the feedback loop of the TIA amplifier, an ambient light filter (developed around the OPA2846) is introduced [89].

In order to reduce the direct current of the photodiode, which occurs due to the luminaires in the underground mine, in the feedback loop of the TIA amplifier, a filter for ambient light (ALF) is introduced, developed around the OPA2846 amplifier. Figure 11 shows the tandem TIA amplifier–light filter used in the oRx.

The following values of the passive elements are necessary according to the system’s requirements and final design: $\;R_{F}=100\;kΩ$; $\;C_{F}=40\;\mathrm{pF}$.

The filter for the ambient light as part of the feedback loop of the TIA is defined by the following values of the passive elements: $\;R_{12}=1\;MΩ$; $\;R_{13}=1.2\;kΩ$; $\;C_{12}=4.7\;\mathrm{nF}$.

The anti-aliasing filter used in the oRx is a low-pass filter with a 2-stage Sallen–Key topology. The electrical diagram of the Sallen–Key filter is presented in Figure 12.

When designing the Sallen–Key filter, we took into account that, for the ATmega328PB microcontroller, the voltage range allowed on the ADC inputs is (0–3.3) Vdc and the maximum resolution is 15 kSPS. For this reason, the cut-off frequency used in the design of the Sallen–Key filter is 90 kHz and the maximum allowable amplification is 18.3 dB.

Following the design, the passive elements obtained for the Sallen–Key filter, are: $\;R_{8}=R_{9}=R_{10}=R_{11}=R_{a}=R_{c}=10\;kΩ$; $\;R_{b}=R_{d}=15.8\;kΩ$;$\;C_{8}=260\;\mathrm{pF}$; $\;C_{9}=420\;\mathrm{pF}$; $\;C_{10}=184\;\mathrm{pF}$; and $\;C_{11}=174\;\mathrm{pF}$. The maximum amplitude obtained after the design is 17.4 dB, obtained at 34.3 kHz. At the cut-off frequency of 90 kHz, the resultant amplitude is 15.4 dB.

The Sallen–Key filter is developed around two LMH6682 [90] amplifiers. The +5 Vdc voltage used in the oRx is obtained with the support of the LT1376HV circuit and the −5 Vdc voltage is obtained with the support of the TPS60403 circuit [91].

### 4.2. The Hybrid Personnel Tracking and Monitoring System (PT&MS)

A hybrid system for PT&M of the atmospheric parameters in underground mines, developed around a star-type topology, based on a portable gas measuring device, is presented in Figure 13. Here, the local wireless communication is performed by VLC technology. The miner’s cap lamps (the VLC oTx) are equipped with various types of transducers.

The proposed PT&MS is made around DKDP PLCs produced by the Polish company Carboautomatyka [92]. The PLC is developed around a 32-bit RISC processor, with a RAM memory of 192 kB and a FLASH memory of 8 MB. The DKDP PLC has an HMI that allows both to display the measured data on a monochrome LCD screen (resolution of 320 × 240), as well as various settings of the PLC through a keyboard. On the other hand, the DKDP PLC also allows the storage of measured data through a storage module in which a maximum of two micro-SD memory cards can be inserted.

These PLCs can be connected both in star topology and in chain topology (Daisy-Chain) by optic fiber or RS485. In the monitoring system shown in Figure 13, the PLCs are connected via optical fiber in a star topology. The number of DKDP PLCs that are connected in the monitoring system in Figure 13 is eight. Each DKDP PLC supports ten measurement lines. On each measuring line, four gas transducers can be connected via RS485, using, on each line, an adapter module, which allows communication both with fixed gas transducers from the SC-xx series [93] and with other compatible devices produced by other companies. Each adapter module also has an analog input, which allows the connection of fixed measuring elements with analog voltage output (0.4–2 Vdc; 1–5 Vdc and 1–10 Vdc). Each adapter module has a DC/DC converter, which raises the voltage to approximately 25 Vdc, thus ensuring the remote supply (maximum 3 km) of the fixed measuring elements.

Each DKDP PLC allows the connection on the DIN rail inside the PLC housing of digital input/output (I/O) modules, on which various digital devices can be connected, as well as various execution elements. Each I/O module has four digital inputs and four digital outputs. Through these digital I/O modules, the related commands can be given to the execution elements within the underground mine. The maximum number of modules (measurement modules and/or digital I/O modules) that can be mounted on the DIN rail in the same housing as the DKDP PLC is ten.

Specifically, the PLCs included in the PT&MS in Figure 13 are DKDP/F1/L8/IO2/K2/W1/A11. This type of PLC has eight adapter modules for data acquisition measured by gas transducers and two digital I/O modules (IO2).

The connection between the fiber optic-based communications network and the network based on VLC technology (the backhaul portion of the network) is made via the DKDP programmable controller, using RS485. Although this wired backhaul solution (via DKDP and RS485) is quite expensive, the solution covers the requirements of underground coal mining, being able to cover approximately 24 km long in main galleries, using DKDP programmable machines controlled in a daisy chain topology. At the ends of the main galleries, as well as in the working faces, the monitoring system can be extended using MX40 programmable controllers, as well as BM25 multi-gas mobile detectors, using a wireless backhaul. The MX40 programmable controller is connected to the DKDP controller using RS485 (see Figure 13). VLC receivers are placed on the main galleries at a long distance from each other, therefore, a handover mechanism in the VLC system is not necessary.

The effect of Doppler (Doppler shift) on LOS can be significantly reduced when the VLC receiver is an intelligent one that has implemented an algorithm for tracking the location of the transmitter (mining lamp). This can be undertaken through a mechanism that allows the rotation of the photodiode in the VLC receiver component, both horizontally and vertically [94].

### 4.3. Validation of the Proposed Communication System through Numerical Simulation in MATLAB-Simulink

#### 4.3.1. Simulation of the Underground Optical Channel of the VLC System

When designing the entire VLC system, the effects of dispersion, attenuation and scattering (that results in signal fading) have to be carefully taken into consideration to avoid a significant limitation of the system’s performance. Therefore, an appropriate front-end design of oTx, as well as proper oRx setup, significantly mitigate the effects of the optical signal fading. Accurate modelling of the optical channel’s characteristics is also critical in this regard.

The VLC setup with LOS topology on the main gallery underground is presented in Figure 14 where the mobile oTx is embedded into the miner’s cap lamp and the fix oRx is embedded into the cabled backbone communication.

The DC gain of oRx in LOS topology for VLC system [95] with a Lambertian source is:(1)$$H_{LOS}\left(0\right)=\left\{\begin{array}{c} \frac{A_{\mathrm{PD}}}{d^{2}}\cdot R\left(\varphi \right)\cdot \cos\left(\omega \right),\;\;\;\;0\leq \omega \leq \omega_{c}\\ \;\;\;\;\;\;\;\;\;\;\;\;\;\;\;\;\;\;\;\;\;\;\;\;0,\;\;\;\;\omega >\omega_{c} \end{array}\right.$$
where: *ω*—incidence angle (rad); *ω*_c_—FOV of the photodiode (rad); *φ*—irradiance angle (rad); A_PD_—radiant sensitive area of the photodiode (m^2^); d—distance between oTx and oRx (m); *R*—Lambertian radiant intensity.
(2)$$R\left(\varphi \right)=\frac{m+1}{2\cdot \pi }\cdot \cos^{m}\left(\varphi \right)$$
where *m* is order of Lambertian emission.
(3)$$m=-\frac{ln\left(2\right)}{ln\left(\cos\left(\varphi_{1/2}\right)\right)}$$
where: $\varphi_{1/2}$—half-power angle (rad)—angle between the emitting direction and the axis, in which luminous intensity is half of the axial intensity [95]. 

The optical intensity received becomes in this situation:(4)$$P_{PD\_LOS}=P_{LED1}\cdot H_{LOS}\left(0\right)$$
where: $P_{LED1}$ is power emitted by LED1 from the mining lamp component (W).

On the other hand, the received diffused power is:(5)$$P_{DIFF}=P_{LED1}\cdot \eta =P_{LED1}\cdot \frac{A_{\mathrm{PD}}}{A_{G}}\cdot \frac{\rho_{1}}{1-\rho }$$
where: $\rho_{1}$—reflectivity of the surface; $A_{G}=2\cdot \left(L\cdot H+L\cdot W+W\cdot H\right)$—mining gallery surface; ρ—average reflectivity; η—the diffuse channel loss.

The total received power is:(6)$$P_{PD\_T}=\left(P_{PD\_LOS}+P_{DIFF}\right)\cdot T_{s}\left(\omega \right)\cdot g\left(\omega \right)$$
where: $T_{s}\left(\omega \right)$—transmission coefficient of the optical filter; $g\left(\omega \right)$—concentrator gain
(7)$$g\left(\omega \right)=\left\{\begin{array}{c} {\left(\frac{n_{c}}{sin\left(\omega \right)}\right)}^{2},\;\;0\leq \omega \leq \omega_{c}\\ \;\;\;\;\;\;\;\;0,\;\;\omega >\omega_{c} \end{array}\right.$$$n_{c}$—index of refraction of the lens (LA1951-A).

Under these conditions, BPX61 photocurrent is: (8)$$I_{P}=P_{PD\_T}\cdot S_{\lambda }$$$S_{\lambda }$—spectral sensitivity of the chip-photodiode (A/W).

The signal-to-noise ratio (SNR), is:(9)$$SNR=\frac{I_{P}^{2}}{\sigma_{T}^{2}}$$
where: $\sigma_{T}^{2}$—is total noise variance.

The total noise variance is:(10)$$\sigma_{T}^{2}=\sigma_{N}^{2}+\sigma_{A}^{2}$$
where:$\;\sigma_{A}^{2}$—amplifier noise variance and $\sigma_{N}^{2}$—shot-noise variance.

The amplifier noise variance is:(11)$$\sigma_{A}^{2}=I_{N}^{2}\cdot G_{B}$$
where: $I_{N}$—input current noise of the OPA2846 amplifier $\left(\frac{A}{\sqrt{Hz}}\right)$; $G_{B}$—amplifier bandwidth (OPA2846).

The shot-noise variance is:(12)$$\sigma_{N}^{2}=2\cdot q\cdot S_{\lambda }\cdot \left(P_{PD\_T}+P_{AN}\right)\cdot G_{N}$$$P_{AN}$—noise power of the ambient light; $P_{AN}=\frac{I_{A}}{S_{\lambda }}$; $I_{A}$—background dark current; $G_{N}=I_{2}\cdot B_{r};$ $I_{2}$—noise-bandwidth factor; $B_{r}$—baud data rate (bps); *q*—elementary electric charge $q=1.60217656535\cdot 10^{-19}\left(C\right)$.

The SNR (dB) is:(13)$$SNR_{dB}=10\cdot \mathrm{lg}\left(SNR\right)$$

In case of an underground VLC setup [96] with a short distance between oTx and oRx, the multipath dispersion is not taken into consideration, a LoS link channel is modelled as a linear attenuation and delay. The system is non-frequency selective and the path loss depends on the square of distance between oTx and oRx. In this case that *φ* < 90°, *ω* < *ω*_c_ and d ≫ $\sqrt{A_{PD\;}}$, the LOS optical channel impulse response (CIR) is given by:(14)$$h_{LOS}\left(t\right)=\frac{A_{P}\left(\omega \right)}{d^{2}}\cdot \frac{m+1}{2\cdot \pi }\cdot \cos^{m}\left(\varphi \right)\cdot \cos\left(\omega \right)\cdot \delta \left(t-\frac{d}{c}\right)$$
where:${\;A}_{P}\left(\omega \right)=A_{\mathrm{PD}}\cdot T_{s}\left(\omega \right)\cdot g\left(\omega \right)$; *c*—speed of the light in vacuum (2.99792458 m/s); δ(.)—Dirac function; δ(t − d/c)—signal propagation delay.

On the other hand, to highlight the propagation losses, we will use the following formula:(15)$$P_{L}=\frac{P_{LED1}}{P_{PD\_T}}$$

In these conditions, the propagation losses in decibels are determined based on the following relationship:(16)$${\left(P_{L}\right)}_{dB}=10\cdot \mathrm{lg}\left(\frac{P_{LED1}}{P_{PD\_T}}\right)$$

Simulation conducted in the MATLAB application (see Figure 15, Figure 16 and Figure 17) regarding the optical channel gain is based on the equations previously presented and characteristics in Table 3.

In Figure 17, it can be seen that the maximum path losses are 51.9 dB (for the distances hx = 0.58 m and d = 1.37 m—in the corners of the space), and the lowest value of the path losses is 49.9 dB. Following the simulation, we can conclude that the path losses increase with the increase of the distance (d) between the transmitter and the receiver. In this sense, path losses can be modeled based on the following relationship [97]:(17)$${\left(P_{L}\right)}_{dB}={\left(P_{L}\left(d_{r}\right)\right)}_{dB}+10\cdot \gamma \cdot \mathrm{lg}\left(\frac{d}{d_{r}}\right)+X_{\sigma }$$
where: ${\left(P_{L}\left(d_{r}\right)\right)}_{dB}$—the path losses at the reference distance $d_{r}$, $X_{\sigma }$—random variable with zero mean and standard deviation σ (the shadowing effect is modeled through this variable), γ—the path loss exponent.

The modeling of VLC channel propagation losses in underground mines, using relationships (17), was studied by J. Wang [97]. J. Wang concluded that the path loss model (in mining roadway), which has the lowest norm of the residual, is given by the following relationship [97]:(18)$${\left(P_{L}\right)}_{dB}=51.84+36.46\cdot \mathrm{lg}\left(\frac{d}{2}\right)+X_{\sigma }$$
where: $d\geq 2\;\left(m\right)$.

The model given by relationship (18) was obtained after a study of the VLC channel based on a certain VLC system and a certain scenario (see [97]). For a distance d = 2, from (18), we obtain that the propagation losses are 51.84 dB (when the shadowing effect is not taken into account). Comparing the path losses in our article with those obtained by J. Wang, we can conclude that they are close. This validates the simulations from our research.

The SNR values in dB related to the miner’s cap lamp position in space are good. When the oTx is positioned under the oRx, the maximum value is 36.3 dB, and in the corners of the space (for the distances hx = 0.58 m and d = 1.37 m), the minimum value is 32.2 dB. From Figure 17, we can conclude that the SNR values increase with the decrease of the distance (d) between the transmitter and the receiver. These results are because of to the relatively low data rate of the single LED embedded into the miner’s cap lamp. According to the results shown in Figure 17, we can conclude that the system designed by us is not limited by the SNR. For the lowest value of the SNR (in dB), the BER obtained is 10^−9^.

The total optical power received (when the optical filter is used) by the oRx has a maximum value of about 5.11 µ*W* (under the oRx) and the minimum value is 3.21 µ*W* in the corners of the space. When the optical filter is not used, total optical power received by the oRx is 5.56 µ*W* and the minimum value is 3.49 µ*W*. On the other hand, as we can see according to Figure 15, at the PD’s output the maximum current is about 3.17 µA when the oTx is positioned under the oRx, and 1.99 µA is the output current in the corners of the space considered. The current emitted by the BPX61 photodiode is transformed into a voltage (*V_IN_*) by means of the TIA amplifier. How $R_{F}=100\left(k\Omega \right)$, we get $V_{IN}=R_{F}\cdot I_{P}$. So, the output voltage of the TIA amplifier has the following maximum and minimum values: ${\left(V_{IN}\right)}_{max}=0.31\;\left(V\right)$ and ${\left(V_{IN}\right)}_{min}=0.19\;\left(V\right)$. The output voltage of the TIA amplifier (*V_IN_*) is amplified and filtered through the Sallen–Key filter (see Figure 12), obtaining at the filter output the voltage (*V_OUT_*). Under these conditions, the output voltage of the Sallen–Key filter has the following maximum and minimum values: ${\left(V_{OUT}\right)}_{max}=2.07\;\left(V\right)$ and ${\left(V_{OUT}\right)}_{min}=1.27\;\left(V\right)$. From those presented, it can be seen that the output voltages of the Sallen–Key filter are included in the (0–3.3) V range, related to the range of the ADCs of the ATmega328PB microcontroller. Therefore, considering the system designed and presented above, the OOK modulation proves to be a proper technique to be used the scenario described, namely the LOS link with a length between the miner’s cap lamp (oTx) and the Access Point (oRx) of about d = 1.37 m. These values can be significantly improved when the oRx is a smart one with a tracking algorithm embedded, aiming to detect the miner’s cap lamp. Another possible solution to increase the LOS link length when the OOK modulation technique is used is to increase the optical reflective value of the surface around the oRx. However, this last solution brings some drawbacks, such as ISI and optical multipath propagation, thus the complexity of the oRx front-end configuration increases. 

Regarding the VLC optical communication channel, as can be seen in Figure 15 (left), the maximum gain is about 2.87 × 10^−6^ and the minimum value is 1.67 × 10^−6^. Hence, the VLC optical CIR has a maximum value of 9.05 × 10^−6^.
The scenario considered above, although an ideal one, can be applied only on the main galleries underground, where the ventilation system is working near the VLC setup, cleaning the air of any impurities. However, the results of the simulation presented in Figure 15, Figure 16 and Figure 17 cannot be applied in the mining working spaces where, due to the mining activity, a high level of dust concentration occurs. The values of optical scattering and absorption of the light on the optical path between LED and PD cannot properly be considered using the existing tools nowadays when we refer to the mining working spaces. The environment of the working spaces underground is filled with suspended tiny particles of coal and rock dust resulting from the mining operation itself. The dimensions, irregular forms of these suspended particles with time-variable of cloud-density, negatively influence the entire VLC system, and consequently, the magnitude of optical power that hits the photodetector’s active area continuously changes.The extinction coefficient (*k*), due to many time-variable parameters underground, cannot be accurately calculated according to equation [98]:
(19)$$k=\lambda \frac{ln\frac{I_{r}}{I_{i}}}{4\pi t}$$
where: *λ*—wavelength; *I_r_*—intensity of light at destination; *I_i_*—intensity of light at source; *t*—medium thickness.

The system complexity is high, the numerical simulations need powerful resources to process data and the time necessary to obtain proper results raises proportionally with the input data considered, namely the extinction coefficient, because of the input characteristics regarding the shape, various dimensions and time-variable concentration of the dust suspended on the direct LOS path between LED and PD.

In the case that the tiny particles are smaller than the light’s wavelength (λ), the particles absorb the incident light and quickly reemit the light in different directions.

When the reemitted light has the same wavelength as the incident light, Rayleigh scattering occurs. When the reemitted light has a longer wavelength (λ) the molecules are left in an excited state and the process is termed Raman scattering. In Raman scattering, secondary photons of longer wavelengths are emitted when the molecule returns to the ground state. The blue wavelength is scattered more than the yellow one. As opposed to sound, the speed of light is slower in water than in air. Molecules of air O_2_ and N_2_ are Rayleigh scatterers for visible light and are more effective at scattering shorter wavelengths (such as blue and violet). Since the underground coal mining environment is predominantly black, without a sunlight source and with low optical power from the artificial light, the Additive White Gaussian noise (AWGN) has a low value, and therefore, interferences from natural light are absent, and are almost negligible from other artificial sources.

Since 2015, there have been a number of significant works on the VLC channel model for underground mines, considering tilt/rotation of the LED/PD, non-flat walls, shadowing and scattering [99,100,101]. 

A good reason why a VLC system is appropriate to be applied in underground mines in the benefit of the dark color of walls, floor and ceiling that absorb light and thus prevent multiple reflections from several surfaces (therefore multipath propagation and inter symbol interference ISI due to multiple bounces of light can be neglected). 

During the mining activity of cutting, the material resulting from exploitation is sprinkled with water; therefore, the Mie scattering has also been considered in the literature [102] since the suspended particles are considered drops of water. The effect of shadowing due to machinery and personnel has also been considered in the literature when the optical channel is modeled for underground mines [103].

Still, in our scenario, because of the narrow width path in underground coal mines where only one worker can walk at a time (is not possible for two workers to walk side by side), shadowing (as well as ISI due to multiple light sources on the same optical path) is also not to be considered.

#### 4.3.2. Presentation of Data Frames and Algorithms Used in Data Transmission and Reception

We use, in our VLC system, the OOK modulation that is considered today the simplest modulation technique that can use Manchester encoding for DC balance, simple clock and data recovery and acceptable BER for wireless visible light data communication. Thus, while transmitting a Manchester encoded message, the light of each data bit is constant, preventing any intra-frame flickering. On the downside, the Manchester code has the disadvantage of larger bandwidth requirements compared to other common codes. 

Although no high data rates are possible to be achieved with this modulation, data necessary to be sent wirelessly by VLC (CH_4_ concentration, the miner’s cap lamp ID and time of measurement) comprise a small amount and the optical link is rather short. More complex modulation techniques (multi carrier modulations), such as DC biased orthogonal frequency-division multiplexing (DCO-OFDM) or Asymmetrically Clipped Optical OFDM (ACO-OFDM), for example, need also more complex front-end devices, therefore both oTx and oRx would be more expensive, which is not the case in the VLC setup that we propose here.

The CH4 concentrations measured with the INIR-ME5% transducer at 4-min intervals is transmitted to a specific VLC receiver based on the OOK modulation. The transmitted data are coded using the Manchester technique to avoid flashing the LED1.

The structure of the data frame transmitted by the mining lamp to a VLC receiver is (see Figure 18):8 bits allocated to the cyclic prefix (CP), necessary for the attenuation of ISI (Inter Symbol Interference);8 bits for the mining lamp ID (each mining lamp has a unique ID, assigned to an employee on a work shift—28 = 256 available IDs);8 bits of data allocated to the measured concentration of CH4 (measurements are made at intervals of 4 min);8 bits for the day (DD) when the CH_4_ concentration was measured (DD/MM/YYYY);8 bits for the month (MM) in which the CH_4_ concentration was measured (DD/MM/YYYY);8 bits for the year (YYYY) in which the CH_4_ concentration was measured (DD/MM/YYYY);8 bits for the time (hh) when the CH_4_ concentration was measured (hh:mm:ss);8 bits for the minute (mm) in which the CH_4_ concentration was measured (hh:mm:ss);8 bits for the second (ss) in which the CH_4_ concentration was measured (hh:mm:ss);4 bits for CRC.

Manchester coding has a logical 0 bit coded through a high–low transition (10), and a logical 1 bit coded through a low–high transition (01). The data transmission algorithm from mining lamps to a specific VLC receiver is presented in Figure 19.

In order to synchronize the mining emitter lamp and the VLC receiver before the transmission of the message, a message consisting of 8 bits (10101010) is transmitted with a bit time of 5 µs. Thus, the synchronization message has a duration of 40 µs. After the transmission of the synchronization message, LED1 is turned off for 2 µs, after which the transmission of the message with the structure in Figure 18 begins. The ATmega328PB microcontroller in the VLC receiver component sequentially reads the synchronization message consisting of the 8 bits with a bit time of 5 µs and if the received message coincides with the transmitted message, it goes to the reception and decoding of the message that has the format of Figure 11. On the other hand, if the transmitted CRC message does not match the CRC calculated by the VLC receiver, the respective message is ignored.

The data reception algorithm from the component of a VLC receiver is presented in the Figure 20.

The structure of the data frame transmitted by the VLC receiver to the PLC is (see Figure 21):

Including in the structure of the data frame related to the MODBUS RTU protocol, the distance between the PLC and the VLCi receiver, in a certain mining gallery (the distance is set by means of the BI/BD buttons and is displayed on a 7-segment display with 3 digits), the approximate position of the worker (who has a unique ID of the lamp) within the workspace can be recorded. On the other hand, based on the unique ID of the mining lamp, the automatic attendance of the worker during the working hours is also performed.

#### 4.3.3. VLC System Simulation

The VLC system to be simulated in MATLAB-Simulink is presented in Figure 22. In the simulation, Manchester encoding is performed by means of an XOR logic gate. Manchester decoding is performed based on a software algorithm. The Manchester decoding algorithm is presented in detail by the Atmel company in the documentation [104]. In the simulation, the VLC communication channel is considered to have an SNR of 32.2 dB (the minimum value for the analyzed scenario—see the Figure 17). In this case, the photodiode output current is Ip = 1.99 µA (within the LED emission—see the Figure 15). This current (Ip) is transformed into a voltage of V_IN_ = 0.19 V by means of the TIA amplifier. On the other hand, at the output of the Sallen–Key filter we obtain a voltage of V_OUT_ = 1.27 V. In the simulation, the bit rate generated by Bernoulli random binary number generator is 1 bits/s. The results obtained after the simulation are presented in Figure 23.

Figure 23 shows that the decoded Manchester code (Manchester oRx) has a delay of 3/4 of the clock period (T = 0.75 s). Following the simulation tests, it was found that good results are obtained even if the VLC communication channel has an SNR of 25 dB. The results obtained through simulation make us believe that the proposed system is a feasible one that can be used in practice.

## 5. Conclusions

Applications of the novel technological discoveries, such as VLC, when are planned to be applied in industries where special rules must be applied, such as underground coal mines, must be developed according to particular regulations that ensure the health and security of personnel and machinery. The ignition of pit gas in coal mines is one of the main triggers of catastrophic events that are “responsible” for numerous losses of life and significant material damage. 

Any device, equipment or machinery designed for coal mines must pass a number of tests that aim to check if characteristics of the radiation emitted by these electrical/optical devices are candidates to ignite the surrounding explosive atmosphere, since the presence of an additional absorber, such as suspended particles, significantly influences the ignition. Therefore, with this article, we designed a novel device with the VLC technology embedded, a device that follows the strict rules according to ATEX Directive 2014/34/EU that covers equipment and protective systems intended for use in potentially explosive atmospheres. 

We propose, in this article, a novel type of the miner’s cap lamp with a double purpose: illuminating the underground spaces and measuring the concentration of CH_4_. The CH_4_ concentrations and the lamp’s ID, as well as the time when data are measured, are local wirelessly transmitted via VLC/Bluetooth technology to the access points (aka oRx) embedded into the remote-cabled communication backbone of the underground mine, fixed on the ceiling of the main galleries. The oRx communicate with the local PLCs via RS485 using the MODBUS RTU protocol.

In the proposed miner’s cap lamp, the CH_4_ concentrations, as well as the time when data are acquired, are stored on an SD memory card, which can be read and formatted at the end of the working program of each miner in a workstation dock situated at the surface of the mine.

The novel miner’s cap lamp emits acoustic and optical signals in case of exceeding the threshold value of the CH_4_ concentration, as risk-warning and at the end of the working schedule as a time-notice.

When data are received from the miner’s cap lamp, the VLC oRx adds to the message transmitted to the local PLC, access point ID (its address, consisting of its approximate position in meters relative to the PLC). Thus, both the position of the worker within the workspace and his identity are approximately determined. Additionally, due to the unique ID of the miner’s cap lamp, the automatic timekeeping of the worker is performed during the working hours.

The entire system must meet the requirements according to the type of intrinsic security protection, category “ia”. Additionally, the battery of the miner’s cap lamp must meet the intrinsic safety requirements “ia”, and the entire oTx front end structure with LEDs must meet requirements of the “op is” optical radiation standard.

In this article, we also present a hybrid system for monitoring atmospheric parameters in the underground mines based on VLC/Bluetooth/RF technology. The architecture of this system is based on a star topology made around 8 DKDP PLCs, which communicates the data to the mine surface, to a process computer. The remote-cabled transmission between the local PLCs and the network switch is performed through optical fiber, using the user datagram protocol. Within this architecture, VLC oRx communicate with DKDP PLCs via RS485 using the MODBUS RTU protocol. Within the proposed monitoring system, MX40 PLCs and BM25 zonal devices are also used. They communicate within a mesh network via RF. MX40 (slave) PLCs communicate with local DKDP (master) PLCs via RS485 using the MODBUS RTU protocol.

The VLC receiver is designed for single-user communication (the VLC receiver communicates with a single mining lamp). The choice of this type of communication was made both for reasons of simplicity and for reasons related to the spatial geometry of the mining gallery.

The VLC communication (between the miner’s cap lamp and the VLC receiver) can be performed at a maximum rate of 115,200 bps. However, the recommended rate is 57,600 bps.

The maximum distance between the DKDP controller and the fixed measuring elements/VLC receivers is a maximum of 3 km. A maximum of four VLC modules can be queried on each telemetry line of the DKDP via RS485. Thus, in the case of the DKDP/F1/L10 programmable controller, 10 × 4 = 40 VLC modules can be interrogated. On the other hand, in the case of the automated device used in the monitoring system from Figure 14 (DKDP/F1/L8/IO2/K2/W1/A11), 8 × 4 = 32 VLC modules can interrogate. If there are more than 32 VLC modules in a mining gallery, the system architecture can be expanded using DKDP/F2/programmable controllers connected to each other via a daisy chain topology, using fiber optics. A maximum of eight DKDP PLCs can be connected in a daisy chain topology. Additionally, for distances less than 1 km, DKDP devices can be connected to each other via RS485, using the EXT_BUS bus. The architecture of the monitoring system is flexible, being able to respond to the high requirements of underground mining. The communication via RS485 between the DKDP PLC and the fixed measuring elements/VLC receivers is 48,000 bps. The DKDP machine interrogates the fixed measuring elements/VLC receivers on a telemetry line at time intervals of maximum 0.25 s.

Currently, in the mining companies in Jiu Valley (Romania) the following underground environmental monitoring systems are in operation:E.M. Lonea: Continuous measuring system—KSP—with 20 measuring points CH_4_;E.M. Livezeni: Continuous measuring system—KSP—with 28 CH_4_ measuring points; CTT63/40U system with 4-min measurement with 10 measurement points;E.M. Vulcan: CTT63/40U system with 4-min measurement with 32 measurement points;E.M. Lupeni: Two CTT63/40U systems with 4-min measurement with 35 measurement points.

Following a detailed analysis of the literature, as well as the current situation in underground mines in the Jiu Valley, several research directions have been identified that can be addressed in the future. These research directions refer to new network topologies between fixed, zonal and personal (portable) devices, as well as the possibility of using new wireless communication channels between these devices. Therefore, zonal devices (such as BM25, Radius BZ1, etc.) can be designed in the future, which communicate with the related receivers (such as those presented in this article) through VLC/LiFi technology.

Due to intense research efforts and novel developed technologies and applications in the optical communication field, a LiFi local communication (a full duplex transmission-VLC upload and IR download, networked transmission with MIMO data transfer) can be considered as a local wireless transmission in coal mines with a high risk of explosion, where the RF communication must be limited or forbidden.

## Figures and Tables

**Figure 1 sensors-23-00692-f001:**
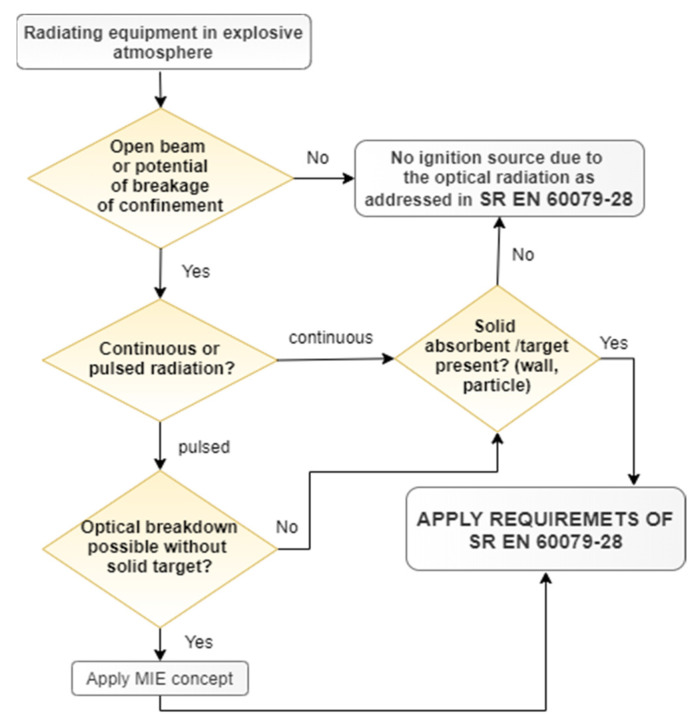
Ignition hazard assessment.

**Figure 2 sensors-23-00692-f002:**
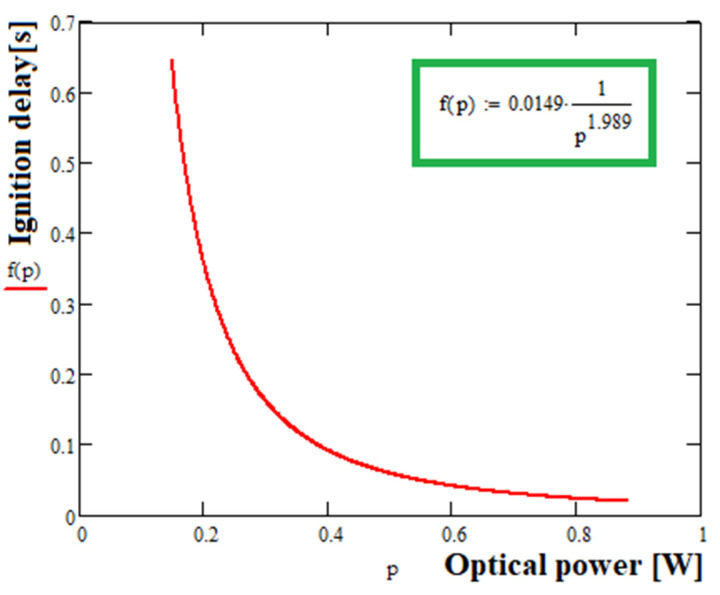
Optical ignition delay times and safe boundary curve with a safety factor of 2 [3].

**Figure 3 sensors-23-00692-f003:**
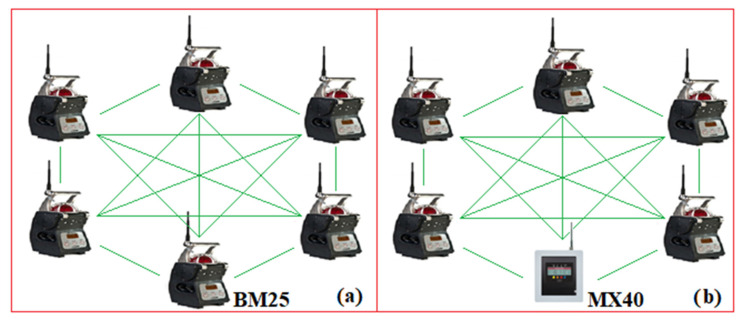
A mesh network (**a**) with six BM25 devices (**b**) with six BM25 devices and a MX40 PLC.

**Figure 4 sensors-23-00692-f004:**
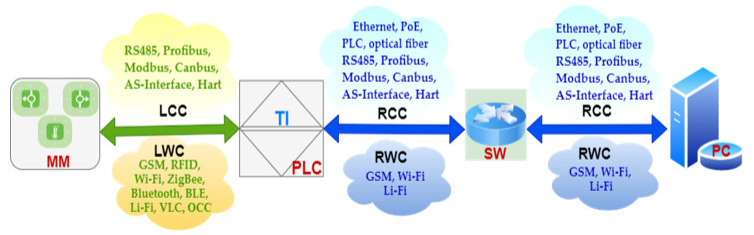
Local and Remote Cabled/Wireless Communication Systems.

**Figure 5 sensors-23-00692-f005:**
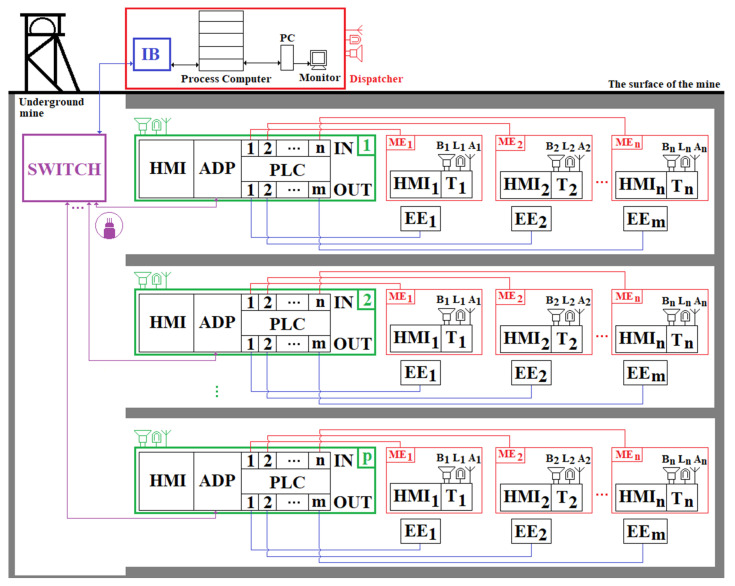
The architecture of the personnel tracking and monitoring system for the underground mine.

**Figure 6 sensors-23-00692-f006:**
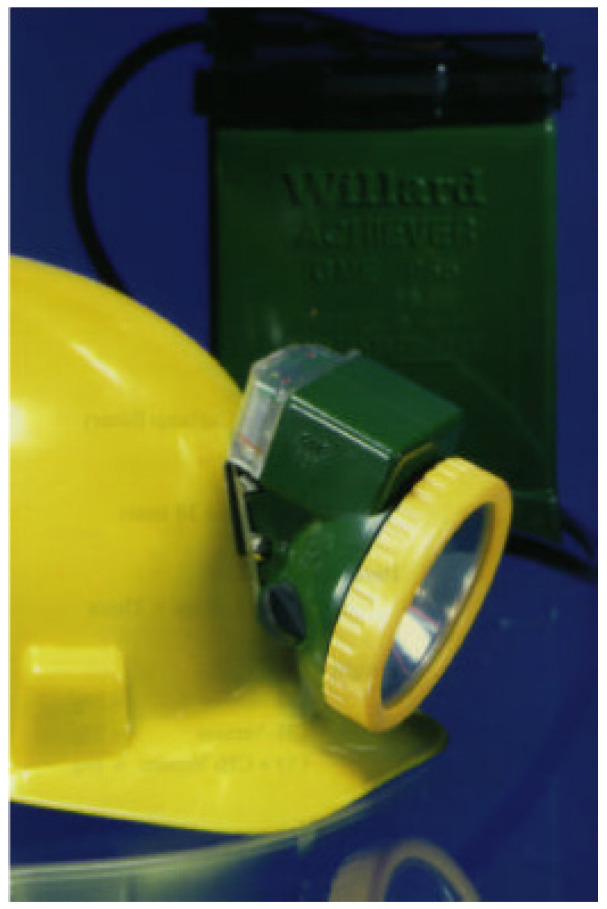
The IYONI II miner’s cap lamp fixed on the helmet.

**Figure 7 sensors-23-00692-f007:**
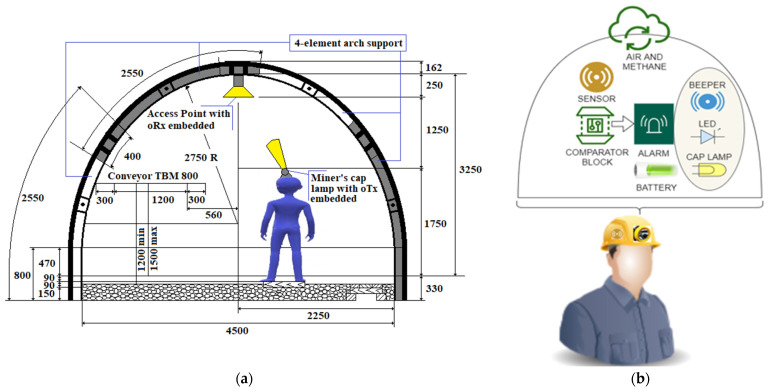
(**a**) The LoS topology between oTx as the miner’s cap lamp fixed on the helmet and the access point as oRx of the VLC setup and (**b**) The component blocks of the miner’s cap lamp, detailed.

**Figure 8 sensors-23-00692-f008:**
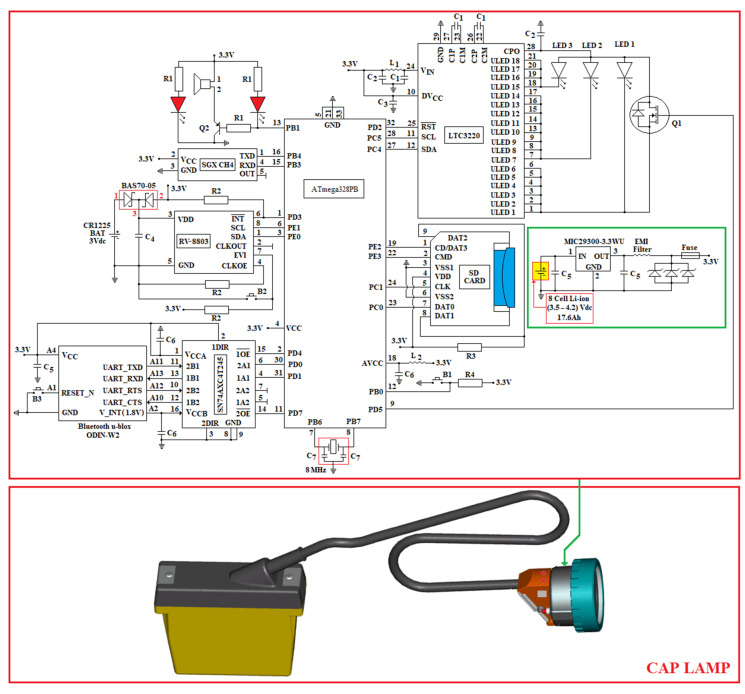
The miner’s cap lamp with wireless VLC technology embedded.

**Figure 9 sensors-23-00692-f009:**
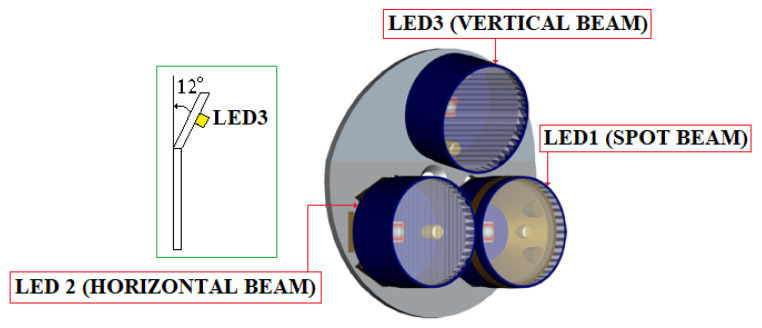
The arrangement of the 3 LEDs within the GEN4-NIOSH lamp headlight [68].

**Figure 10 sensors-23-00692-f010:**
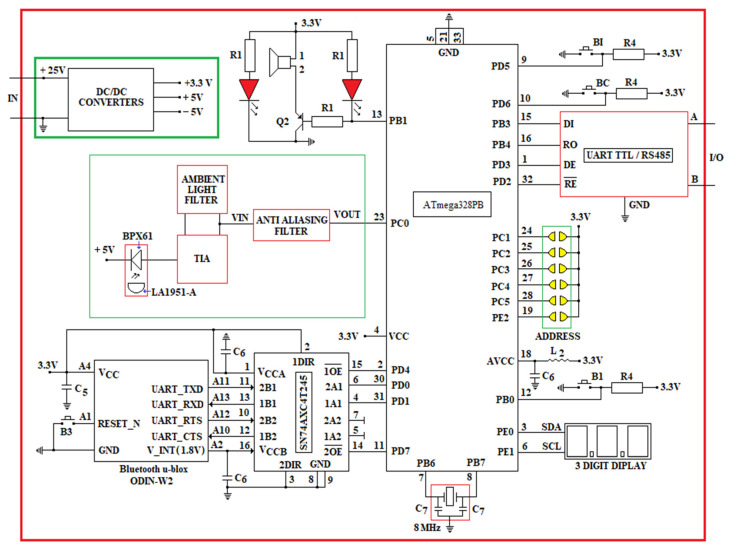
The oRx front end device aka the Access Point.

**Figure 11 sensors-23-00692-f011:**
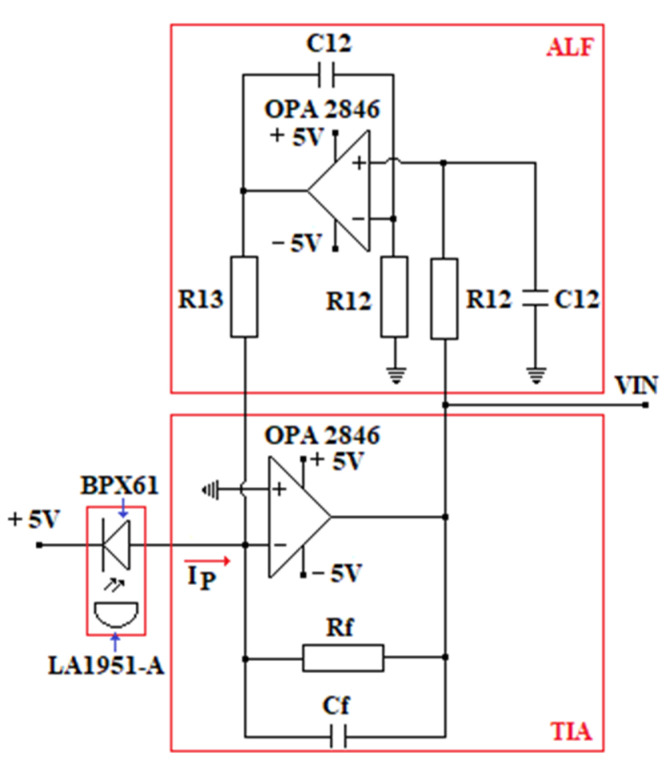
The electronic diagram of the TIA in tandem with the ALF filter.

**Figure 12 sensors-23-00692-f012:**
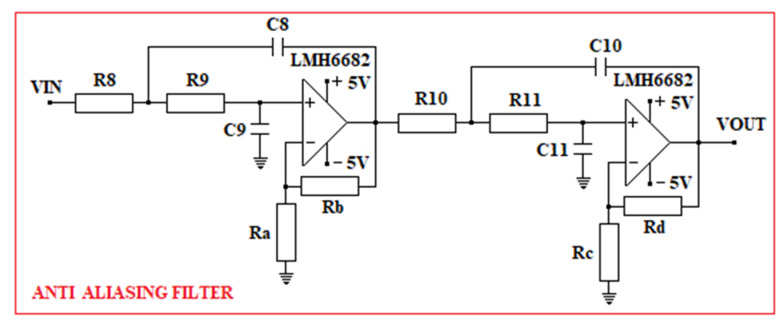
The Sallen–Key filter passes down in 2 steps.

**Figure 13 sensors-23-00692-f013:**
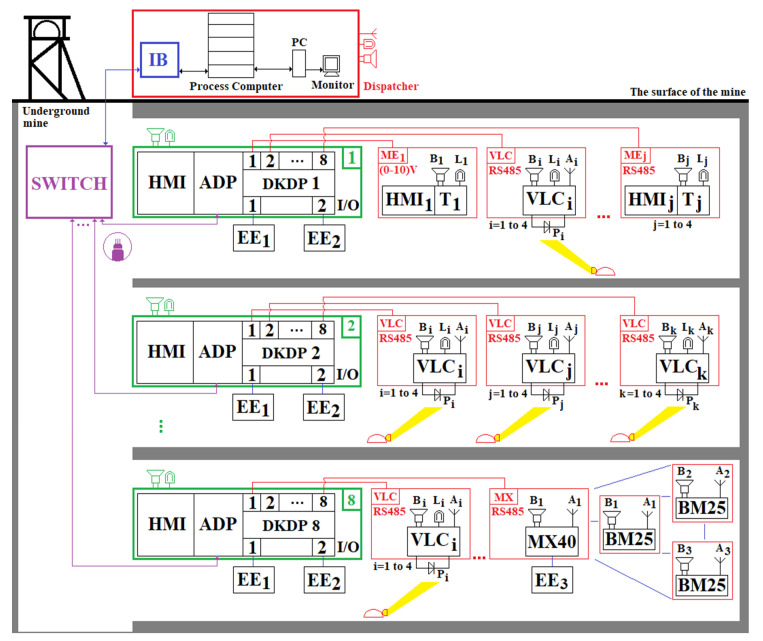
Hybrid personnel tracking and monitoring system of underground mining parameters based on VLC technology.

**Figure 14 sensors-23-00692-f014:**
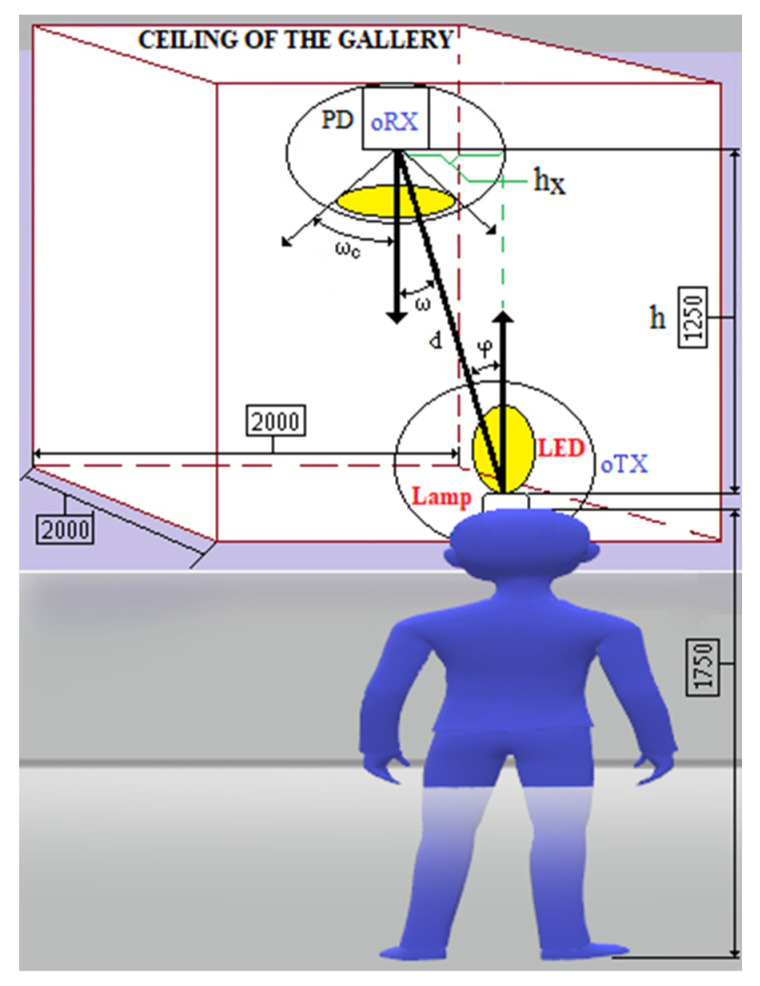
The VLC setup with LOS topology on the main gallery underground.

**Figure 15 sensors-23-00692-f015:**
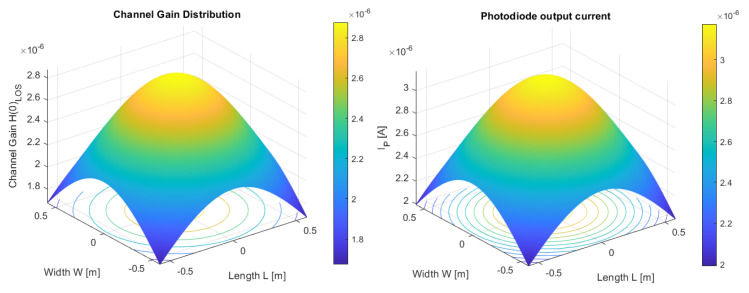
Channel gain in a LoS topology (**left**); Photodiode current distribution (**right**).

**Figure 16 sensors-23-00692-f016:**
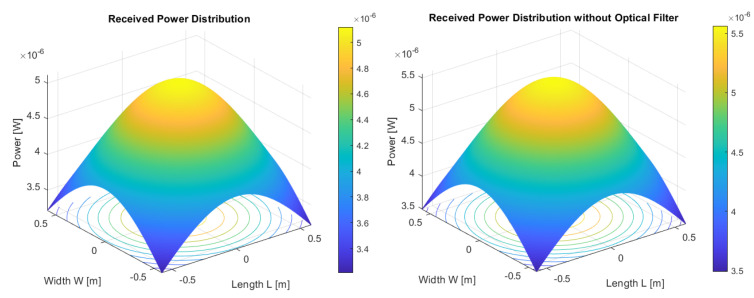
Total received power distribution with optical filter (**left**); Total received power distribution without optical filter (**right**).

**Figure 17 sensors-23-00692-f017:**
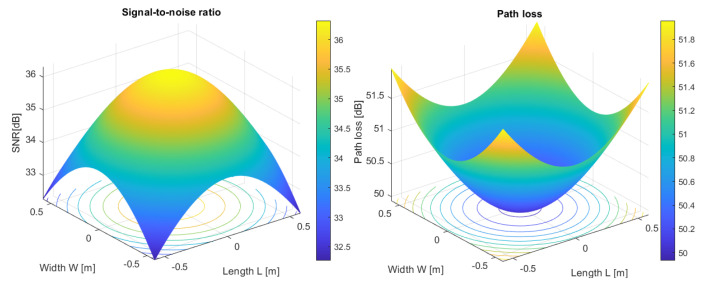
Signal-to-noise ratio (SNR) distribution in dB (**left**); Path loss distribution in dB (**right**).

**Figure 18 sensors-23-00692-f018:**
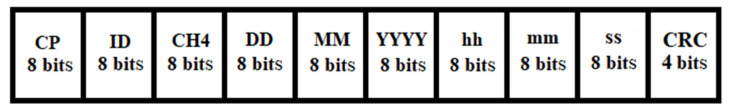
The data frame that is send by the miner’s cap lamp.

**Figure 19 sensors-23-00692-f019:**
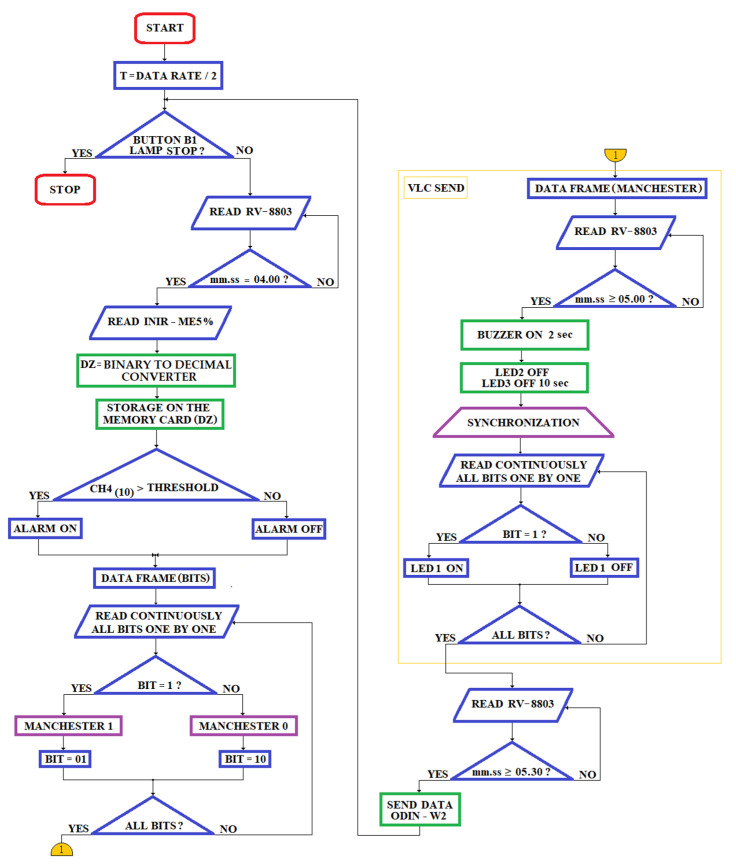
The data transmission algorithm within the mining lamps.

**Figure 20 sensors-23-00692-f020:**
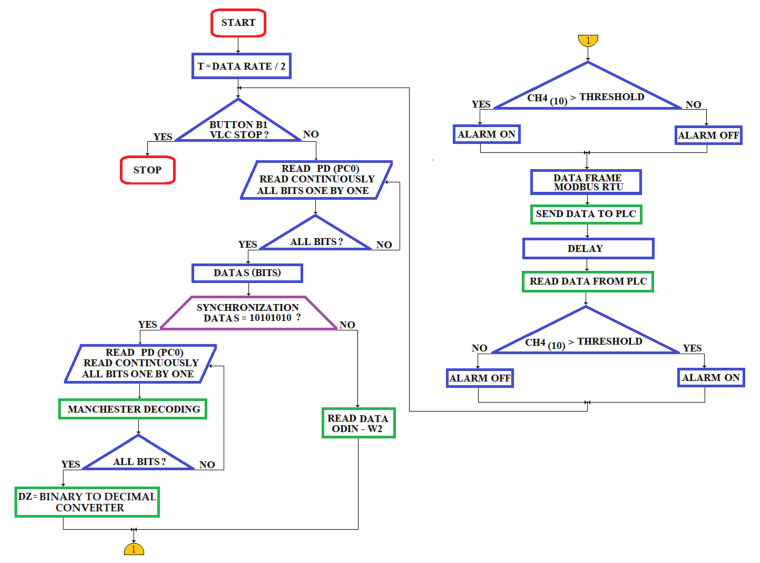
The data reception algorithm of a VLC receiver.

**Figure 21 sensors-23-00692-f021:**
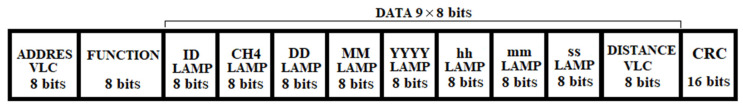
The structure of the MODBUS RTU protocol data frame.

**Figure 22 sensors-23-00692-f022:**
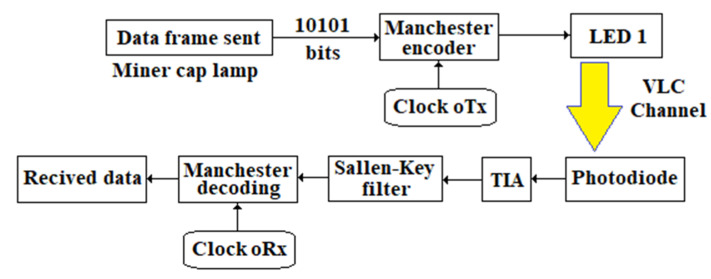
Block diagram of the VLC communication system.

**Figure 23 sensors-23-00692-f023:**
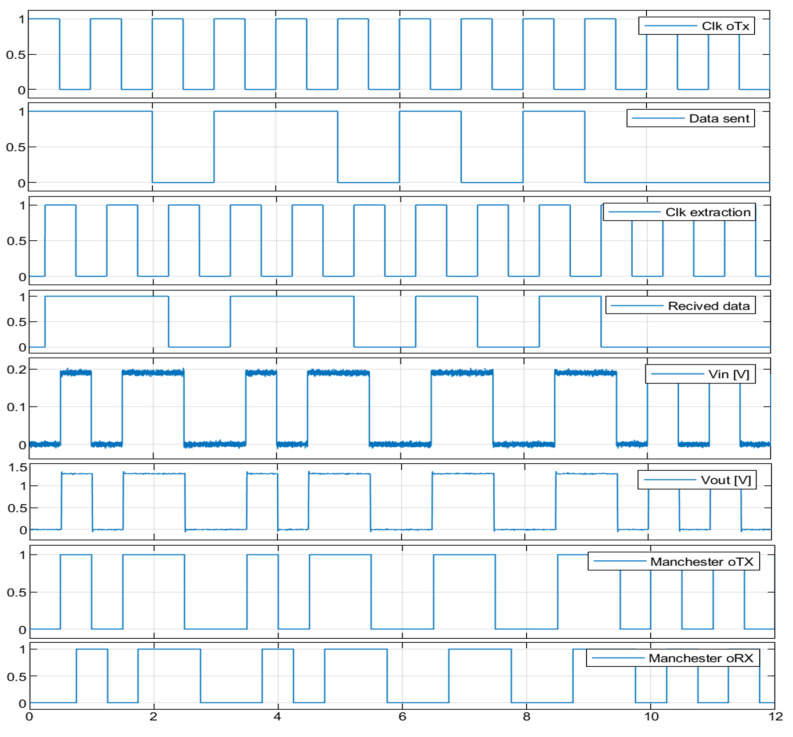
The VLC system simulation results.

**Table 1 sensors-23-00692-t001:** Safe optical power and irradiance for Group I and II equipment, categorized by Equipment Group and temperature class [3].

Optical Radiation Sources with		Can Be Used in the Following Cases	Remarks
Radiated Power (No Irradiance Limit Applies) mW	Irradiance (No Radiated Power Limit Applies) mW/mm^2^		
≤150	-	IIA with T1, T2 or T3 and I	No limit to the involved irradiated area (IIA)
≤35	-	IIA, IIB independent of T-Class, IIC with T1, T2, T3 or T4 and I	No limit to the IIA
≤15	-	All atmospheres	No limit to the IIA
-	≤20	IA with T1, T2 or T3 and I	Irradiated areas limited to ≤30 mm^2^
-	≤5	All atmospheres	No limit to the IIA

**Table 2 sensors-23-00692-t002:** Safe limit values for intermediate area, Group I or II, constant power, T1–T4 atmospheres, equipment Groups IIA, IIB or IIC [3].

Limited Irradiated Area	Maximum Radiated Power Value
<4 ×10^−3^	35
≥4 × 10^−3^	40
≥1.8 × 10^−2^	52
≥0.2	60
≥0.8	80
≥2.9	100
≥8	115
≥70	200
	400
For irradiated areas equal to or above 130 mm^2^ the irradiance limit of 5 mW/mm^2^ applies	

**Table 3 sensors-23-00692-t003:** Characteristics considered for channel model underground.

Nr.	Characteristic	Symbol	M.U.	Value
1	Irradiance semi-angle	*φ*	rad.	25·π/180
2	Transmitted optical power by LED	*P_LED1_*	W	0.504
3	Radiant sensitive area of the BPX61	*A_PD_*	m^2^	7.02 × 10^−6^
4	Height between LED and PD	*h*	m	1.25
5	Dimensions of space considered	*L × W × H*	m	2 × 2 × 3
6	FOV of the BPX61	$\omega_{c}$	rad.	55·π/180
7	Reflectivity of the surface	$\rho_{1}$	-	0.7
8	Average reflectivity	ρ	-	0.5625
9	Transmission coefficient of the optical filter	$T_{s}$	-	0.92
10	Index of refraction of the LA1951-A	$n_{c}$	-	1.515
11	Spectral sensitivity of the chip-BPX61	$S_{\lambda }$	A/W	0.62
12	Input current noise-OPA2846	$I_{N}$	$A/\sqrt{Hz}$	2.8 × 10^−12^
13	OPA2846 amplifier bandwidth (TIA)	$G_{B}$	Hz	300 × 10^6^
14	Background dark current	$I_{A}$	A	30 × 10^−9^
15	Noise-bandwidth factor	$I_{2}$	-	0.562
16	Baud data rate	$B_{r}$	bps	57,600

## Data Availability

Not applicable.

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
