# Peer review of "Application of Optical Communication for an Enhanced Health and Safety System in Underground Mine"

_sensors, 2023, doi:10.3390/s23020692_

Round 1

Reviewer 1 Report

1. What path loss models are considered for the optical portion of the communication?

2. Getting an SNR of 100dB is practically impossible. What is the significance of the figure that plots SNR?

3. Received power is very less. How sensitive is the photo detector to capture this power and react?

4. Are reflection, refraction, scattering and diffraction considered while computing the received power?

5. What is the performance criteria to assess the VLC system developed?

6. How it can be effective during mishaps during the mining process?

7. Does it provide any mechanism to detect land slides inside mines to prevent loss of lives?

8. What is the time taken by the system to report a mishap? How action is taken there after?

Author Response

Dear reviewer,

Thank you for your questions and suggestions.

The answers are in the attachment of this email.

Best regard,

Assoc. Prof. Dr. Eng. Olimpiu Stoicuta

Reviewer 2 Report

The Authors have presented a work on " Application of Optical Communication for an  Enhanced Health and Safety System in Underground Mine". The topic is very relevant to underground mine safety applications. The Authors have proposed the use of visible light communication for local-wireless transmission and the optical fiber for the remote-cabled transmission. Following are the observations which may be addressed;

1. The limitations of VLC communication are well known. The Authors may highlight, how they are going to address the same in their work.

2. how the Hybrid RF/VLC is going to work seamlessly in mine envirnoment? 

3. How the cross-sensitivity issue due to multiple transducers is being addressed. 

4. For the VLC simulation the Authors have used Manchester encoding. The Authors may highlight the reason behind the selection of Manchester encoding/decoding. 

Author Response

(The authors gave the same response as above.)
